rsob.royalsocietypublishing.org

## Perspective

 

**Subject Area:**
microbiology

emergence, evolution, phylogeny, virus, spill-over, virosphere

**Author for correspondence:**
Edward C. Holmes
e-mail: edward.holmes@sydney.edu.au

# Predicting virus emergence amid evolutionary noise

Jemma L. Geoghegan[1] and Edward C. Holmes[2]

[1]Department of Biological Sciences, Macquarie University, Sydney, New South Wales 2109, Australia
[2]Marie Bashir Institute for Infectious Diseases and Biosecurity, Charles Perkins Centre, School of Life and Environmental Sciences and Sydney Medical School, The University of Sydney, Sydney, New South Wales 2006, Australia

ECH, 0000-0001-9596-3552

The study of virus disease emergence, whether it can be predicted and how it might be prevented, has become a major research topic in biomedicine. Here we show that efforts to predict disease emergence commonly conflate fundamentally different evolutionary and epidemiological time scales, and are likely to fail because of the enormous number of unsampled viruses that could conceivably emerge in humans. Although we know much about the patterns and processes of virus evolution on evolutionary time scales as depicted in family-scale phylogenetic trees, these data have little predictive power to reveal the short-term microevolutionary processes that underpin cross-species transmission and emergence. Truly understanding disease emergence therefore requires a new mechanistic and integrated view of the factors that allow or prevent viruses spreading in novel hosts. We present such a view, suggesting that both ecological and genetic aspects of virus emergence can be placed within a simple population genetic framework, which in turn highlights the importance of host population size and density in determining whether emergence will be successful. Despite this framework, we conclude that a more practical solution to preventing and containing the successful emergence of new diseases entails ongoing virological surveillance at the human–animal interface and regions of ecological disturbance.

> Prediction is very difficult, especially about the future.
>
> —Niels Bohr

## 1. Introduction

Emerging infectious diseases have the potential to wreak havoc on agricultural industries and native flora and fauna, and can pose a significant challenge to the health and economic status of both developed and developing countries. The broad-scale drivers of the apparent increase in the number of emerging diseases are well documented, involving such factors as climate change, environmental disruption, an increasingly centralized agricultural system and rapid global transportation, as well as high densities of humans, animals and crops. Combined, these factors have created new opportunities for viruses and other pathogens to change their host range and cause epidemic disease. In humans, this confluence of factors has led to the emergence of a number of high-profile viral infections over the last 40 years, including the ongoing global epidemic of human immunodeficiency virus (HIV), pandemic influenza A viruses such as H1N1/09, SARS coronavirus (CoV), MERS-CoV, and the more recent outbreaks of Ebola virus in West Africa and Zika virus throughout the Americas.

Importantly, the study of disease emergence has moved from making lists of emerging diseases and the proximate ecological factors responsible for

their appearance (e.g. see [1]), to establishing more rigorous quantitative frameworks to explain how new infections become established in populations (e.g. [2–4]). In particular, there is now a large body of literature devoted to understanding the determinants of disease emergence, with a strong focus on emerging viruses. This work has the implicit and commendable goal of trying to reveal the overarching rules of disease emergence, which in turn might lead to predictions of what virus might emerge next and where this may occur [5–8]. Gaining this sort of predictive capability would have obvious and wide-ranging benefits. In these approaches, the study of virus emergence is often synonymous with the study of virus evolution, such that the more we understand about the patterns and processes of evolutionary change, the more accurate any emergence prediction is thought to be.

However, accurate predictions of disease emergence must be based on a correct and rigorous understanding of how viruses jump between hosts and adapt to new transmission cycles, including the time scale over which these processes occur. We show here that a more meaningful understanding of virus emergence requires us to shift the focus away from the broader processes of virus evolution and towards the short-term factors that influence the probability of the successful establishment of a virus in a host population. In other words, if the goal is to develop a meaningful predictive model of disease emergence, there may be considerable value in tuning out the background 'noise' of virus evolution rather than building the model around long-term evolutionary processes. More fundamentally, we will argue that a more practical approach to the challenge of virus emergence will involve abandoning prediction in favour of genomic surveillance at the ecological 'fault-lines' of emergence.

## 2. The nature of virus emergence

The successful emergence of a virus in a new host will often entail a significant adaptive challenge. Indeed, one of the most important observations in disease emergence is that not all viruses that jump species boundaries successfully evolve onward transmission in the new host. Rather, many such viruses appear as transient 'spill-over' infections that soon die out, even in the absence of infection control. For example, despite repeated spill-over events from birds to humans, H5N1 avian influenza virus has not been able to evolve sustained human-to-human transmission [9]. Other viruses have been more successful and resulted in significant outbreaks in new hosts. For example, Ebola virus (EBOV) has caused several localized epidemics that have been largely restricted in their spread by administrative boundaries and border closings [10]. Hence, although there is evidence of the active adaptation of EBOV to humans during the recent 2013–2016 outbreak in West Africa [11,12], the virus clearly possesses the base-line virological traits needed to ensure its onward transmission in the new host. Finally, some viruses have evolved to become endemic human pathogens, involving the generation of well-established and long-standing chains of transmission that do not require repeated spill-over events from an animal reservoir. An obvious case in point is HIV, the agent of AIDS, although a wide variety of human viruses fall into this class. Indeed, it is likely that most endemic virus infections in humans ultimately resulted

from cross-species transmission, although in the majority of cases the exact animal reservoir species are unknown or unsampled. For example, although hepaciviruses are being increasingly documented in animal populations, it is likely that the true reservoir species for human hepatitis C virus has yet to be identified [13,14]. These different virus–host associations, from spill-over to endemicity, highlight the two central questions in the evolution of virus emergence: (i) Why are only some viruses able to successfully spread in a new host? (ii) What barriers, both ecological and genetic, prevent active host adaptation from taking place?

There has been a great deal of experimental research in a number of systems directed towards identifying those specific viral genomic mutations responsible for successful host adaptation [15], although as noted above an important limitation is that there are still relatively few cases in which the precise chain of evolutionary events from reservoir to recipient species have been determined [16]. As expected, many mutations that promote successful host adaptation are concerned with aspects of virus-receptor binding [11,12,17], although changes in other traits, such as pH [18] and interactions with host antiviral responses [19,20], are also of importance. However, virus genetics alone cannot explain why only some emerging viruses are successful. Indeed, even viruses that appear to be well adapted to a specific host (i.e. that seemingly harbour all necessary host-specific mutations) may fail to spread.

An informative example concerns the recent emergence of the A/H3N8 subtype of canine influenza virus (CIV). Although this virus was first recorded in dogs in the USA in the early 2000s, with horses acting as the reservoir host [21], it has failed to become established in the domestic dog population. Instead, CIV is largely confined to dog shelters, where most dogs are infected soon after they arrive [22]. CIV clearly possesses all the genetic characteristics necessary to spread in dogs, and its reproductive number in dog shelters is always sufficient (i.e. $R_0 > 1$) to allow its spread within these confined spaces. However, CIV has failed to ignite a wider epidemic in dogs, probably because contact heterogeneity in the domestic dog population is much greater than in dog shelters such that there is an insufficient density of susceptible hosts for the outbreak to take hold [22]. This inhibition of virus emergence through a lack of susceptible hosts is likely to be commonplace. The general lesson to be learned for exercises in prediction is that determining whether a virus can spread in a particular host, for example following cell passage experiments or using animal models, does not mean that it will in the real world unless epidemiological circumstances are permissive. Virus emergence should therefore always be thought of as a combination of successful genetics aligned with permissive ecology [4,8,16].

## 3. Is emergence predictable?

Predicting emergence has become one of the highest stakes topics in the study of infectious disease. The multi-host dynamics of virus emergence, from reservoir to recipient hosts, requires us to consider the interplay of host ecology and virus genetics. At the ecological level, emergence risk has been associated with such factors as climate change, population demographics [1], geographical 'hotspots' of former emergence locations [5] and host plasticity [23]. At

rsob.royalsocietypublishing.org   Open Biol. 7: 170189

the virus genetic level, there has been effort to determine those virological factors that correlate with transmissibility within a specific host [7] (see below). Theoretical studies aimed at assessing the predictability of emergence have strived to encompass the complexities of inter- and intra-host evolutionary fitness dynamics [2,24–26], where the within-host and between-host fitness landscapes play a central role in determining the probability of emergence [4,24].

A central goal of research in this area has been to reveal the 'rules' that underpin disease emergence, on the implicit assumption that predictive accuracy will follow. A more ambitious scheme was established in 2016 in the guise of the Global Virome Project (GVP). Through a global partnership the GVP aims to identify and characterize 99% of zoonotic viruses with epidemic potential to better predict, prevent and respond to future viral threats [27]. To achieve these aims, the GVP will perform large-scale metagenomic surveys of viruses in vertebrate populations. The underlying logic is that knowing what viruses are present in nature provides substantive value in trying to determine what will come next. The GVP is therefore the clarion call for studies in predicting virus emergence.

Surveys of virus biodiversity have already revealed remarkable levels of genomic and phylogenetic diversity [28,29], and will undoubtedly increase our understanding of the patterns and processes of virus evolution. However, we contend that they are unlikely to be informative in predicting the next pandemic. Most obviously, the total number of viruses on earth—the virosphere—is so large that it can never provide a guided insight into what viruses may eventually emerge in humans. There are an estimated 8.7 million eukaryotic species on earth [30]. If we assume that each carries approximately 10 species-specific viruses (more than 200 viruses have been documented in humans), then we can very roughly estimate that there are in the order of 87 million eukaryotic viruses perhaps distinct enough to be considered different species. Currently, the International Committee on Taxonomy of Viruses (ICTV) recognizes 4404 virus species in all hosts (both eukaryotes and prokaryotes), which means that 99.9949379% of the eukaryotic virosphere remains undiscovered or unclassified (figure 1), and these calculations ignore the huge numbers of bacteriophage.

The idea of a virosphere so expansive is supported by the vast numbers of viruses discovered by recent studies of viral biodiversity that have been stimulated by advances in metagenomics, particularly the use of bulk RNA sequencing [28,29,31]. Importantly, these metagenomics studies have considered virus diversity in terrestrial species, whereas previous studies had a strong focus on aquatic environments and DNA bacteriophage [32–34]. Most dramatically, a recent metagenomic analysis of nine invertebrate phyla identified 1445 novel RNA viruses, as well as newly defined genera and families (and possibly orders) [28]. Not only does this represent a major increase in our knowledge of virus diversity, but that it came from a survey of only 220 species from a small number of sampling locations in China hints at the true scale of the virosphere.

Although vertebrates, particularly mammals, may carry a smaller number of viruses, the number is still so very large as to make any detailed experimental follow-up of even the vertebrate virosphere impractical, particularly as the rapid nature of RNA virus evolution means that any individual

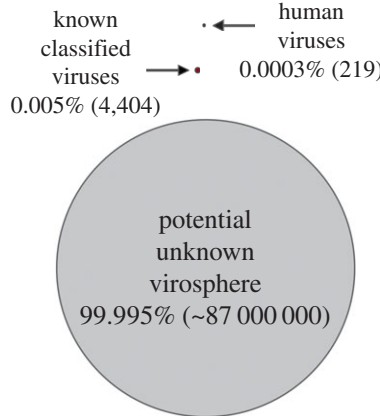

**Figure 1.** Illustration of the relative size of the potentially unknown virosphere. The estimates shown are based on the assumption that approximately 10 viruses might be capable of infecting each of the estimated 8.7 million eukaryotic species on earth [26]. Currently, the ICTV recognizes 4404 virus species in both eukaryotes and prokaryotes (although many others are awaiting classification), which means that 99.9949379% of the virosphere remains undiscovered or unclassified.

virus species will harbour a wide diversity of ever-changing variants. This impracticality is augmented when one considers our lack of knowledge of whether this vast set of viruses can replicate in human cells, and even this trait will not guarantee that a virus will be able to successfully transmit between hosts. In this context it has been proposed that machine learning may help in pandemic prediction, for instance by using sequence data to predict which cell receptors a virus might use [35,36]. However, attaining knowledge of cell receptor compatibility in itself does not enable accurate predictions of emergence, particularly as viruses with a diverse range of receptors are able to infect humans. For example, it has long been known that influenza viruses bind to sialic acid-containing molecules as receptors [37]. However, this information has not improved prediction of influenza virus emergence and re-emergence. More generally, machine learning requires very large amounts of data to predict common events, whereas studies of disease emergence necessarily use data on *rare* events to predict *rare* events.

Paradoxically, then, the more we sample animal populations, the less frequently virus cross-species transmission to humans seems to occur. For example, when SARS coronavirus (CoV) was revealed to have its origin in bats [38], the total number of known bat viruses was so very small that the likelihood that a bat virus might emerge in humans correspondingly appeared to be relatively high. However, the total number of known bat viruses has increased dramatically with better sampling [6,39,40], and bat-to-human zoonotic transmission now appears to be a rare event.

It is also important to recall that the most recent viruses to achieve epidemic spread in humans—Ebola and Zika—were caused by known and well described human pathogens, with the first descriptions of Zika virus going back to the 1940s [41]. Yet our previous knowledge of these pathogens was not indicative of their epidemic potential. It may therefore be the case that the greatest pandemic threat in fact lies in those viruses that re-emerge intermittently in large and dense host populations. We propose a theoretical framework to understand this possibility later.

rsob.royalsocietypublishing.org Open Biol. 7: 170189

rsob.royalsocietypublishing.org    Open Biol. 7: 170189

**Figure 2.** Possible effects of host biodiversity on the probability of viral emergence. The red arrows at the bottom depict instances of increased emergence risk. Wildlife host species richness has been proposed as an important predictor of disease emergence. Likewise, host populations of low biodiversity might harbour fewer viruses and a decreased risk of emergence. Conversely, high host biodiversity has also been linked to a decrease in disease risk through the 'dilution effect'.

Previous attempts to predict aspects of virus emergence have met with limited success. There has been considerable interest in trying to predict the geographical locations where viruses may emerge in the future, based on identifying the locations where such emergence events have occurred in the past—the so-called 'hotspots' of emergence [5]. Although these may well represent localities where sampling virus biodiversity will be profitable, it is difficult to turn such studies into a viable index of predictability. For example, both Mexico and Saudi Arabia appear as 'cold' spots in these emergence maps, with relatively little evidence of past disease emergence. Yet, since these hotspot maps were published, Mexico has witnessed the emergence of H1N1/09 virus (in pig populations), while MERS coronavirus emerged in Saudi Arabia in 2012, with dromedary camels unexpectedly acting as the reservoir host [42].

It has also been suggested that wildlife host species richness is an important predictor of disease emergence [5]. Conversely, however, biodiversity has also been linked to a decrease in disease risk through the 'dilution effect' [43–47]. This was first developed as a framework to infer the dynamics of tick-borne Lyme disease, and describes the association between increasing species richness and reduced disease risk, particularly when the most competent hosts were dominant in the community and alternative hosts negatively influenced the dominant hosts as reservoirs [43,48]. Although still debated, the dilution effect highlights the central role of host biodiversity and ecology in shaping the epidemiology of disease-causing pathogens. Inevitably, habitat destruction and ecosystem disturbance due to changes in land use will contribute to the loss of biodiversity. The broader consequences of such losses for emerging human pathogens are unknown and clearly merit further investigation (figure 2).

Although cross-species virus transmission sits at the heart of virus emergence, phylogenetic studies of the frequency with which different virus families are able to jump species boundaries also offer little predictive power as all exhibit a strong tendency to jump hosts. Indeed, it now appears that the evolutionary history of most virus families comprises a complex mix of cross-species transmission and virus–host

co-divergence, and that trying to disentangle the respective contributions of each process will be challenging [49]. In addition, the greater diversity of hosts and their viruses sampled, the more cases of species jumping we are likely to document [49]. Importantly, these phylogenetic studies also demonstrate that virus–host associations, including cross-transmission, may extend over many millions of years, and not only in the recent past, as is assumed in studies of virus emergence. As a case in point, it is possible that the Narna–Levi group of RNA viruses have co-diverged with their hosts since the α-proteobacteria became endosymbionts [28].

While other comparative analyses have revealed those virological factors that increase the transmissibility of emerging viruses in humans [7], these analyses also probably offer little predictive power. These studies suggest that viruses with low host mortality, that establish chronic infections, that are non-segmented, that do not possess an envelope and that are not transmitted by vectors have greater 'emergibility' in humans [7]. Nonetheless, many viruses still fall into this class and a number of these traits are not measurable until the virus has already established itself in a new host, diminishing the predictive utility of such 'viral traits' analyses.

Given these uncertainties, and the fact that elements of the evolutionary process that underpins emergence are inherently unpredictable, we suggest that there is no simple algorithm that will enable an accurate prediction of what viruses might emerge in the future. Hence, it is necessary to lower our expectations about disease emergence as a predictive science. In particular, although metagenomics undoubtedly has major implications for our understanding of virus evolution, it also probably undermines biodiversity-based attempts to predict the virus source of the next major disease pandemic [6]. There are clearly so many viruses in nature that trying to determine which will ultimately appear in a new host from diversity sampling alone is almost certainly futile.

Predictions also sit uneasily with most aspects of evolutionary biology. Even relatively simple traits like virulence, which have generated considerable evolutionary

rsob.royalsocietypublishing.org Open Biol. **7**: 170189

theory, have proven difficult to predict because of myriad unknown forces that shape their evolutionary trajectory [50]. Although there has been some success in using phylogenetic approaches to predict the short-term evolution of human influenza virus [51], the nature of the central selective processes shaping virus evolution (i.e. antigenic drift) is well known and to some extent quantifiable over the time scale studied. This is demonstrably not the case when considering unknown emerging viruses.

## 4. The conflation of epidemiological and evolutionary time scales

At face value it seems obvious that evolutionary ideas and analyses will help predict the emergence threat posed by different viruses. However, a major limitation is that evolutionary processes, particularly those reliant on phylogenetic or other comparative analyses, often occur on a markedly different time scale than the epidemiological processes relevant to pandemic prediction. Indeed, one of the most important conclusions of recent work in the study of RNA virus evolution is that the time scale over which these viruses have evolved, including cross-species transmission events, is probably far longer than previously imagined. This realization comes from both phylogenetic studies of virus biodiversity and branching patterns [49,52], particularly the match between parts of the virus and host trees, and the analyses of endogenous virus elements that act as genomic fossils [53]. Hence, it is likely that many of the viral families that infect vertebrates have done so for many millions of years, and have experienced continual cross-species transmission since this time.

Although central to understanding evolutionary processes, these time scales are irrelevant for predicting the next pandemic within an epidemiological time scale (i.e. 1–10 years). The same caveat applies to studies that have used the taxonomic span covered by viral families as a way of determining which have the greatest propensity to jump hosts [6]. These taxonomic ranges may have taken millions of years to generate and not the scale of years necessary for effective pandemic prediction. Evolutionary and epidemiological time scales should not simply be assumed to be equivalent. Although phylogenies can be used to accurately describe both macro- and micro-evolution, and superficially appear similar, the trees at these two scales are produced by markedly different evolutionary processes (figure 3). As it is clear that the pace of human ecological (anthropogenic) change generally occurs more rapidly than successful virus host-jumping depicted in a phylogenetic tree, from a public health point of view we would do better to monitor ongoing environmental disturbance by humans than quantify long-term aspects of virus evolution.

An informative example of this fundamental disconnect between evolutionary and epidemiological time scales is provided by the hepadnaviruses, which include human hepatitis B virus (HBV). There is strong evidence for hepadnavirus–host co-divergence stretching back for effectively the entire time-span of vertebrate evolution [52,54]. Cross-species transmission has occurred on this background of co-divergence, with a recent analysis revealing approximately 13 host jumps over an evolutionary period of approximately 400 million years [49]. Although our sample of hepadnaviruses is inevitably small, with new hepadnaviruses

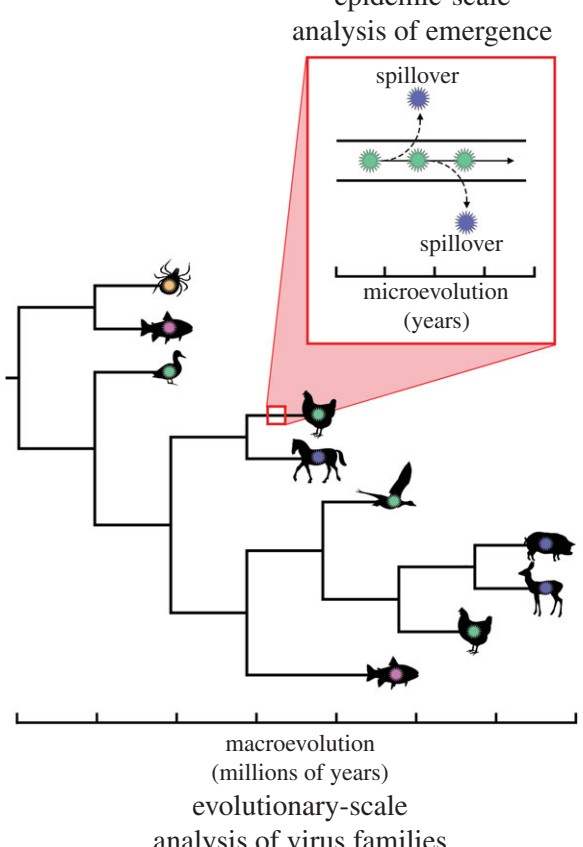

**Figure 3.** Phylogenetic analyses of a virus family, seemingly showing many instances of cross-species transmission over evolutionary time scales (i.e. virus macroevolution). Critically, however, the adaptive processes (i.e. mutation and selection) that lead to virus 'spill-overs' and possible emergence in a new host are more informative when considering a shorter, microevolutionary time scale.

recently identified in fish [54], and more cases of cross-species transmission will assuredly be found, this very roughly equates to a successful cross-species transmission event every 30 million years. Even if the rate of host jumping is 10 000 times more frequent, occurring once every 3000 years, this is still far too broad a time scale to provide any meaningful predictive value for the study of human disease emergence. A similar story can be told for the influenza viruses. Although these are exemplars of cross-species transmission [55], which occurs frequently in the *Orthomyxoviridae* [49], it is still problematic to make these predictions over the time scale of human observation. For example, the emergence of H3N8 equine influenza virus from an avian host took place in the early 1960s. Although this virus is clearly adapted for mammalian respiratory transmission, there is no evidence that it has transmitted to humans during the last 50 years.

Those cases in which viruses have been deliberately released as biological controls also highlight the disconnect between evolutionary and epidemiological time scales. These natural experiments proceed over epidemiological time scales which in many ways parallel the natural emergence and spread of a novel virus in a new host. Most notably, both myxomavirus (a poxvirus) and rabbit haemorrhagic disease virus (a calicivirus) have been successfully released as biological controls into populations of European rabbits in Australia and Europe, in the 1950s and 1990s,

rsob.royalsocietypublishing.org    Open Biol. 7: 170189

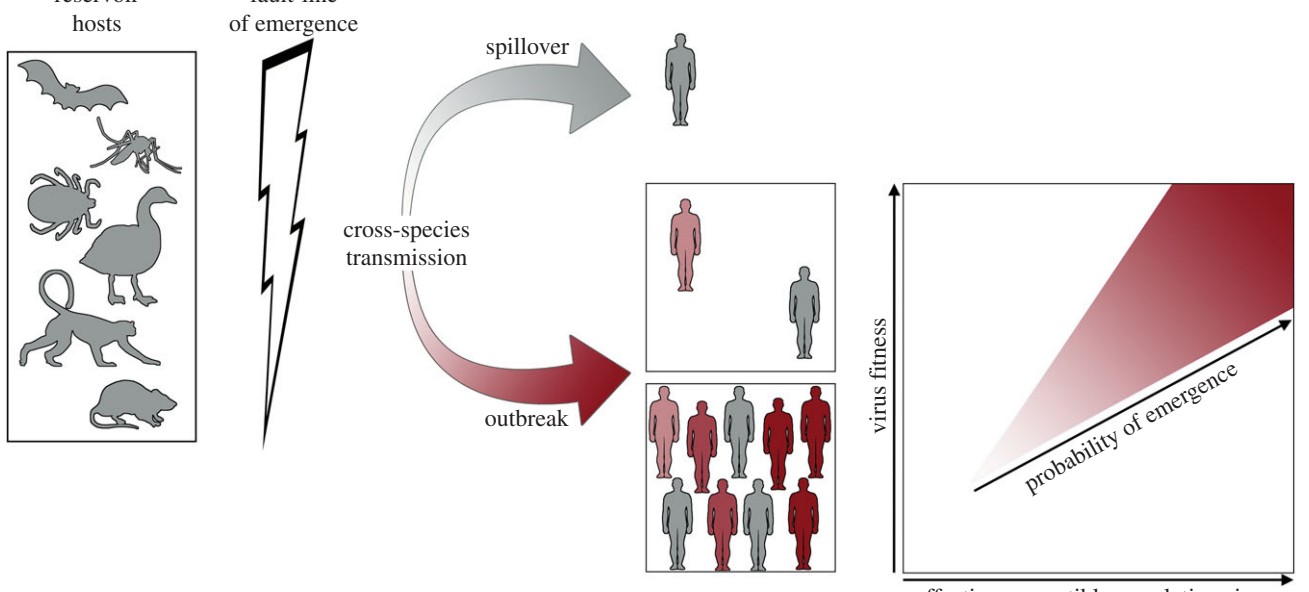

**Figure 4.** Exemplifying the human–animal interface as the fault-line of disease emergence. Following a cross-species transmission event, a virus might cause a dead end 'spill-over' or it might be genetically adapted to be transmissible between members of the new host species. Even for emergent viruses of high fitness the probability of emergence and the size of the outbreak relies on a large and dense host population, as well as a variety of other ecological factors that can be thought of as comprising the 'effective susceptible population size' (x-axis, right-hand panel).

respectively [56]. What is particularly striking is that to date there are no strongly supported cases of these viruses jumping into other (i.e. non-lagomorph) species over the time scale of release, even though both virus families appear to experience very frequent host-jumping over long evolutionary time scales [49]. Hence, the observation that poxviruses can frequently jump species boundaries over evolutionary time scales provides no assistance in predicting what happens on the shorter time scale that govern epidemics.

## 5. A population-genetic framework to understand virus emergence

The study of virus emergence represents a synthesis of two different types of scientific enquiry: virology, which aims to determine, usually experimentally, the mutations that enable a virus to infect a new host; and epidemiology, which primarily seeks to identify the ecological factors responsible for viruses crossing the species boundary and spreading in a new host.

We contend that these approaches can be synthesized within a single population genetic framework. Specifically, the cross-species transmission and emergence of a virus in a new host can be envisioned as a simple form of the adaptive process, in which the subject under consideration is the acquisition of mutations that facilitate replication and transmission and hence increase viral fitness. Although they may be of myriad form, the ecological factors that dictate whether such an emergence event will be successful are directly analogous to the random sampling effects that necessarily impact the spread of any new allele in a population, increasing the likelihood of genetic drift that will in turn result in stochastic loss. For example, the extensive contact heterogeneity (i.e. lack of susceptible hosts) that prevents CIV spreading in the domestic dog population is equivalent to

the fate of an advantageous allele in a small host population. That is, although the virus (mutation) may be host-adapted (advantageous), it will not spread far because the host population is so small/sparse that genetic drift dominates substitution dynamics (and even strongly advantageous alleles may be lost rather than fixed in small populations).

We suggest that this new population genetic view of the process of cross-species transmission and emergence can be achieved by making the move away from thinking about viruses spreading horizontally through a population (by host-to-host transmission), which is the realm of epidemiology, and towards thinking about virus alleles/genes being inherited vertically, which is the domain of population genetics. A well-understood framework is that successful cross-species transmission requires three steps: (i) encounter a new host species, (ii) infection of that new host and (iii) propagation in the new host population [57]. Adaptation to the new host species may often represent a major challenge, as mutations that are beneficial in this host also probably decrease fitness in the reservoir host species. This opposing selection between reservoir and recipient hosts shapes the adaptive landscape of viral emergence [24]. For example, as the gradient of the adaptive landscape increases, genetic variants are subjected to stronger opposing selection between the reservoir and the recipient hosts. Models of this adaptive process therefore offer an indication of in which part of parameter space the host adaptation of a novel virus might be possible [2,4,24,26].

Importantly, however, this adaptive process must also occur within the background of random sampling. Because of a lack of host contacts, or descendants, genetic drift will reduce sampling of the fittest virus and decrease the probability of emergence. Hence, as the host population increases in size, the probability that a virus will be sampled increases (figure 4), although it is likely that additional factors, including prior immunity and population age

rsob.royalsocietypublishing.org Open Biol. 7: 170189

structure, will also impact the probability of virus sampling. A simple lesson from this new approach is that host population size and density—which can be thought of as comprising the 'effective susceptible population size' for an emerging virus—will have a major impact on whether a new virus will successfully spread in a population, irrespective of the fitness of a particular mutation (i.e. whether a virus contains all the mutations necessary to adapt it to a new host). Consequently, if the fitness of a virus and the effective susceptible population size were known, or even measurable, it would be possible to make bounded estimates of how likely a successful emergence event might be.

The importance of genetic drift can also be seen in the transmission bottlenecks that will routinely occur as a virus moves between hosts [58], which probably put a brake on host adaptation [58–62]. Even if a specific variant is favoured within an individual host, but does not increase sufficiently in frequency (i.e. such that is still found at sub-consensus levels), then a severe population bottleneck may result in its loss. Clearly, the more severe the population bottleneck, the less natural selection will be able to optimize viral fitness at the epidemiological scale. Despite the insights provided by this population genetic framework, the vastness of the unknown virosphere means that all such theories of emergence are probably of more use in predicting the population impact of emergence rather than predicting what might emerge next and in what location.

# 6. Sampling the human–animal interface: fault-lines of disease emergence

If the accurate prediction of virus emergence is impractical, or even unattainable, what can be done to help prevent the emergence of major epidemic or pandemic disease in humans? Cross-species transmission events highlight the public health threat of wildlife trade and consumption, and heightened contact between animal hosts and humans has probably facilitated these events. It is therefore critical to direct our attention to the animal–human interface (humans, livestock trade and consumption, wildlife, environment) as this can be thought of as the 'fault-line' at which most disease emergence events occur.

We therefore propose that a more effective practical strategy for managing emerging and re-emerging epidemic or pandemic disease is the targeted surveillance of viromes at the human–animal interface. The vast biodiversity of viruses in the animal world makes their analysis prior to any emergence in humans a Sisyphean exercise. Rather, humans are the best sentinels: a virus discovered in humans very obviously can replicate in that host, which will not be the case for myriad viruses identified through biodiversity surveys of other animal taxa.

We therefore urge regular genomic surveillance at the fault-line of disease emergence that captures this human–animal interface (figure 4). Examples of this interface that could be sampled are those associated with (i) major changes in land-use (particularly human encroachment into forest areas during deforestation), (ii) occupational exposure to live animal markets, and (iii) changes in human demographics, behaviour and political instability that result in population mobility and displacement. To take one specific example, the hunting and butchering of wild animals, and the meat trade that flows from it, is common practice among many countries. This activity must represent a conduit for cross-species pathogen transmission, and is probably responsible for the transfer of simian retroviruses from infected non-human primates to humans [63]. Virological surveillance of those working in the bushmeat trade therefore appears a necessary measure. Importantly, accelerating environmental and anthropogenic changes are expanding the human–animal interface [58], and the rapid movement of humans and livestock, as well as agricultural produce, highlights the importance of effective surveillance.

This virome surveillance should be ongoing and performed simultaneously on multiple human populations globally, with existing serological data perhaps helping to determine which geographical locations harbour human populations most frequently exposed to animal viruses, and hence where virome surveillance will be most informative. In addition, while metagenomics is hugely powerful in characterizing the viromes of individual organisms, including the discovery of new species, it requires active infection (replication) and that samples be taken from tissues that contain the virus. For this reason meta-serological surveys will also be of importance as they enable the identification of infections that have occurred in the recent past.

# 7. Conclusion

Predicting virus emergence has risen to become a key goal of the study of infectious disease. The study of virus evolution has revealed much about the nature of virus emergence and its history over evolutionary time scales. However, due to the fundamental differences between evolutionary and epidemiological time scales, a focus on virus evolution may in fact be a distraction when it comes to predicting the next virus pandemic. Similarly, while virological features that increase the likelihood of virus emergibility can be identified, these features cannot be treated as hard and fast rules determining which viruses will successfully emerge. Further, many of these features are only capable of being observed *after* emergence occurs, such that they are likely to be of little predictive power. In partial response to these problems, we suggest that the field may be advanced by using a population genetic framework that melds genetic and ecological studies of virus emergence, and which highlights how the effective susceptible population size of a new host plays a major role in dictating the chance of successful emergence. In this manner we identify the possibility of a meaningful theoretical framework for the study of emergence that is grounded in evolutionary theory, but that tunes out the 'noise' of virus macroevolution.

Despite such a framework, the inconvenient truth for all those working in the realm of disease emergence is that the vastness of the unknown virosphere and the diverse range of viruses that have achieved endemic transmission in humans means that any attempt to predict what virus may emerge next will face substantial, and probably crippling, difficulties. In light of this we suggest it may be of more benefit to public health to target, via surveillance, the fault-line of disease emergence that is the human–animal interface, particularly those shaped by ecological disturbance. Once a virus is identified as being of interest in this manner, other analyses may be able to assess its impact and pandemic potential. Such a shift in focus, away from being able to

make predictions of emergence based on fundamental rules and towards the better assessment of emergence impact, is both more achievable and more likely to provide positive public health outcomes.

Data accessibility. This article has no additional data.

Authors' contributions. J.L.G. and E.C.H. jointly conceived and wrote the paper. Both authors gave final approval for publication.
Competing interests. We declare we have no competing interests.
Funding. E.C.H. is funded by an Australia Fellowship from the National Health and Medical Research Council, Australia (GNT1037231).

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
