## [Reviewer comments · Open Biology]

Review History

RSOB-17-0189.R0 (Original submission)

Review form: Reviewer 1

Recommendation

Accept with minor revision (please list in comments)

Are each of the following suitable for general readers?

- a) **Title**
Yes
- b) **Summary**
Yes
- c) **Introduction**
Yes

Is the length of the paper justified?

Yes

Should the paper be seen by a specialist statistical reviewer?

No

Is it clear how to make all supporting data available?

Not Applicable

Is the supplementary material necessary; and if so is it adequate and clear?

Not Applicable

Do you have any ethical concerns with this paper?

No

Comments to the Author

I tremendously enjoyed reading this article, which is beautifully written. I could not agree more with the argument that prediction of viral emergence is inherently difficult (although I am an optimistic and would not go as far as calling this a 'futile' exercise). I fully concur with the authors that cataloguing viral diversity, as in the Global Virome Project, is overpromising our forecasting ability.

I have only a few minor comments:

- The authors provide nice examples of emergence 'failures', whereby viruses that seemingly have the required genetic makeup for successful transmission in a new host still fail to cause large-scale outbreaks, presumably due to ecological issues (the poster child being Canine Influenza Virus, CIV, here). There is however very little in this paper on the viruses that successfully caused full blown epidemics (HIV, SARS, pandemic flu, and perhaps a few more examples from non-human diseases systems). It would be useful to elaborate on these 'success' stories for contrast, summarize the processes of genetic adaptation (if any) and the permissive ecological conditions present during the emergence phase.

- Using CIV as a case study, the authors highlight the importance of permissive ecological factors, which they summarize as "host population size" in Fig 4. However, it is more complex than sheer population size. As a case in point, the global canine population size is huge and would likely be able to sustain CIV outbreaks if it were a more connected population. Further, other factors beyond population structure could hinder viral emergence, including prior immunity and age structure (an emergent virus may be immunologically related to other viruses already encountered by a new host population, eg flu or enteroviruses). Perhaps a better terminology for what the authors mean is 'effective susceptible population', which could encompass population size and turn-over, contact networks and age structure, prior immunity, and probably other factors. A discussion of these ecological factors would be useful somewhere in the text.

- In thinking about the 'fault-lines' of disease emergence at the animal-human interface, can the authors elaborate on the geographic regions where sampling should be prioritized? Clearly, there has been much attention devoted to identifying hotspots of viral diversity and regions associated with rapid ecological changes. This broadly aligns active animal-human interfaces, where I think the authors would like to see increased sampling. However, these fault lines fail to explain the emergence of the 2009 pandemic virus, MERS-CoV, or Zika virus. Do we need more research on identifying these fault lines in a more quantitative way, and if so, could sampling of viral diversity across species, and more behavioral/ecological data on species diversity and contacts, help in any way? Or are we left with sampling already recognized hotspots, which may help in only a fraction of viral emergence situations?

Decision letter (RSOB-17-0189)

12-Sep-2017

Dear Dr Holmes,

We are pleased to inform you that your manuscript RSOB-17-0189 entitled "Predicting virus emergence amidst evolutionary noise" has been accepted by the Editor for publication in Open Biology. The reviewer(s) have recommended publication, but also suggest some minor revisions to your manuscript. Therefore, we invite you to respond to the reviewer(s)' comments and revise your manuscript.

Please submit the revised version of your manuscript within 14 days. If you do not think you will be able to meet this date please let us know immediately and we can extend this deadline for you.

- 1) A text file of the manuscript (doc, txt, rtf or tex), including the references, tables (including captions) and figure captions. Please remove any tracked changes from the text before submission. PDF files are not an accepted format for the "Main Document".
- 2) A separate electronic file of each figure (tiff, EPS or print-quality PDF preferred). The format should be produced directly from original creation package, or original software format. Please note that PowerPoint files are not accepted.
- 3) Electronic supplementary material: this should be contained in a separate file from the main text and meet our ESM criteria (see <http://royalsocietypublishing.org/instructions-authors#question5>). All supplementary materials accompanying an accepted article will be treated as in their final form. They will be published alongside the paper on the journal website and posted on the online figshare repository. Files on figshare will be made available approximately one week before the accompanying article so that the supplementary material can be attributed a unique DOI.

Online supplementary material will also carry the title and description provided during submission, so please ensure these are accurate and informative. Note that the Royal Society will not edit or typeset supplementary material and it will be hosted as provided. Please ensure that the supplementary material includes the paper details (authors, title, journal name, article DOI). Your article DOI will be 10.1098/rsob.2016[last 4 digits of e.g. 10.1098/rsob.20160049].

4) A media summary: a short non-technical summary (up to 100 words) of the key findings/importance of your manuscript. Please try to write in simple English, avoid jargon, explain the importance of the topic, outline the main implications and describe why this topic is newsworthy.

Images

Data-Sharing

It is a condition of publication that data supporting your paper are made available. Data should be made available either in the electronic supplementary material or through an appropriate repository. Details of how to access data should be included in your paper. Please see <http://royalsocietypublishing.org/site/authors/policy.xhtml#question6> for more details.

Data accessibility section

Sincerely,
The Open Biology Team
<mailto:openbiology@royalsociety.org>

Author's Response to Decision Letter for (RSOB-170189)

See Appendices A & B.

RSOB-17-0189.R1 (Revision)

Review form: Reviewer 1

Recommendation

Accept as is

Are each of the following suitable for general readers?

- a) **Title**
Yes
- b) **Summary**
Yes
- c) **Introduction**
Yes

Is the length of the paper justified?

Yes

Should the paper be seen by a specialist statistical reviewer?

No

Is it clear how to make all supporting data available?

Not Applicable

Is the supplementary material necessary; and if so is it adequate and clear?

Not Applicable

Do you have any ethical concerns with this paper?

No

Comments to the Author

The paper was nicely revised - no further comment.

Decision letter (RSOB-17-0189.R1)

25-Sep-2017

Dear Dr Holmes

We are pleased to inform you that your manuscript entitled "Predicting virus emergence amidst evolutionary noise" has been accepted by the Editor for publication in Open Biology.

If applicable, please find the referee comments below. No further changes are recommended.

You can expect to receive a proof of your article from our Production office within approx. 5 working days. Please let us know if you are likely to be away from e-mail contact during this period. Due to rapid publication and an extremely tight schedule, if comments are not received, we may publish the paper as it stands.

Sincerely,

The Open Biology Team

mailto: openbiology@royalsociety.org

As a conscientious publisher, Open Biology is keen to get your opinion on the publishing system so we can adapt and make the process more author-friendly. In order to achieve this, we would like to invite you to participate in a survey being conducted by Editage Insights by clicking on the following link: <https://www.surveymonkey.com/r/author-perspectives-on-academic-publishing-royal-society>

This should take no more than 15 minutes and you will have the opportunity to enter a prize draw. We hope these results will provide us with valuable insights we can use to improve our service.

Appendix A

Referee: 1

Comments to the Author(s)

I tremendously enjoyed reading this article, which is beautifully written. I could not agree more with the argument that prediction of viral emergence is inherently difficult (although I am an optimistic and would not go as far as calling this a 'futile' exercise). I fully concur with the authors that cataloguing viral diversity, as in the Global Virome Project, is overpromising our forecasting ability.

Response: We thank the reviewer for these positive comments.

I have only a few minor comments:

- The authors provide nice examples of emergence 'failures', whereby viruses that seemingly have the required genetic makeup for successful transmission in a new host still fail to cause large-scale outbreaks, presumably due to ecological issues (the poster child being Canine Influenza Virus, CIV, here). There is however very little in this paper on the viruses that successfully caused full blown epidemics (HIV, SARS, pandemic flu, and perhaps a few more examples from non-human diseases systems). It would be useful to elaborate on these 'success' stories for contrast, summarize the processes of genetic adaptation (if any) and the permissive ecological conditions present during the emergence phase.

Response: The reviewer makes a fair point. We have therefore revised the paper to include some extra details on these virus 'success stories', focusing on HIV and HCV, and with a number of additional references. We also note that most (if not all) endemic viral infections likely have an animal reservoir, even though we have not identified that reservoir in most cases (with HCV providing a good example). Hence, for most of the success stories we don't actually know what mutations have been involved in human (or non-human) adaptation. We now make this point in the paper.

- Using CIV as a case study, the authors highlight the importance of permissive ecological factors, which they summarize as "host population size" in Fig 4. However, it is more complex than sheer population size. As a case in point, the global canine population size is huge and would likely be able to sustain CIV outbreaks if it were a more connected population. Further, other factors beyond population structure could hinder viral emergence, including prior immunity and age structure (an emergent virus may be immunologically related to other viruses already encountered by a new host population, eg flu or enteroviruses). Perhaps a better terminology for what the authors mean is 'effective susceptible population', which could encompass population size and turn-over, contact networks and age structure, prior immunity, and probably other factors. A discussion of these ecological factors would be useful somewhere in the text.

Response: We agree with the reviewer and have revised the paper accordingly. By 'host population size' we were really thinking of a combination of host population size AND density. We have now clarified this in the paper. However, the reviewer is clearly correct that we need more nuance here and hope we have provided this in the revised version of the paper. Indeed, we now explicitly mention prior immunity and age structure. We really like the idea of 'effective susceptible population size' – sums it up nicely – and have therefore revised Figure 4 and the associated text accordingly.

- In thinking about the 'fault-lines' of disease emergence at the animal-human interface, can the authors elaborate on the geographic regions where sampling should be prioritized? Clearly, there has been much attention devoted to identifying hotspots of viral diversity and regions associated with rapid ecological changes. This broadly aligns active animal-human interfaces, where I think the authors would like to see increased sampling. However, these fault lines fail to explain the emergence of the 2009 pandemic virus, MERS-CoV, or Zika virus. Do we need more research on identifying these fault lines in a more quantitative way, and if so, could sampling of viral diversity across species, and more behavioral/ecological data on species diversity and contacts, help in any way? Or are we left with sampling already recognized hotspots, which may help in only a fraction of viral emergence situations?

Response: This is a good question. It seems counter-productive to list precise geographic locations given our general nervousness about predictions. However, we have now suggested that existing serological data might be a good way of identifying those human populations that are commonly exposed to animal pathogens and hence where sampling might be most profitable.

Appendix B

Predicting virus emergence amidst evolutionary noise

Jemma L. Geoghegan¹ and Edward C. Holmes²

¹Department of Biological Sciences, Macquarie University, Sydney, NSW 2109, Australia

²Marie Bashir Institute for Infectious Diseases and Biosecurity, Charles Perkins Centre, School of Life and Environmental Sciences and Sydney Medical School, The University of Sydney, Sydney, NSW 2006, Australia

Author for correspondence:

Professor Edward C Holmes,

Marie Bashir Institute for Infectious Diseases and Biosecurity, Charles Perkins Centre,

School of Life and Environmental Sciences and Sydney Medical School,

The University of Sydney, Sydney, NSW 2006, Australia.

Email: edward.holmes@sydney.edu.au

The study of virus disease emergence, whether it can be predicted and how it might be prevented, has become a major research topic in biomedicine. ~~Herein~~Here we show that efforts to ~~actively~~ predict disease emergence commonly conflate fundamentally different evolutionary and epidemiological timescales, and are likely to fail because of the enormous number of unsampled viruses that could conceivably emerge in humans. Although we know much about the patterns and processes of virus evolution on evolutionary timescales as depicted in family-scale phylogenetic trees, these data have little predictive power to reveal the short-term microevolutionary processes that underpin cross-species transmission and emergence. Truly understanding disease emergence therefore requires a new mechanistic and integrated view of the factors that allow or prevent viruses ~~to spread~~spreading in ~~host populations~~novel hosts. We present such a view, suggesting that both ecological and genetic aspects of virus emergence can be placed within a simple population genetic framework, which in turn highlights the importance of host population size and density in determining whether emergence will be successful. Despite this framework, we conclude that a more practical solution to preventing and containing the successful emergence of new diseases entails ongoing virological surveillance at the human-animal interface and regions of ecological disturbance.

Keywords

Emergence; Evolution; Phylogeny; Virus; Spill-over; Virosphere

'Prediction is very difficult, especially about the future'

Niels Bohr

1. Introduction

Emerging infectious diseases have the potential to wreak havoc on agricultural industries and native flora and fauna, and can pose a significant challenge to the health and economic status of both developed and developing countries. The broad-scale drivers of the apparent increase in the number of emerging diseases are well documented, involving such factors as climate change, environmental disruption, an increasingly centralised agricultural system, rapid global transportation, as well as high densities of humans, animals and crops. Combined, these factors have created new opportunities for viruses and other pathogens to change their host range and cause epidemic disease. In humans, this confluence of factors has led to the emergence of a number of high-profile viral infections over the last 40 years, including the ongoing global epidemic of human immunodeficiency virus (HIV), pandemic influenza A viruses such as H1N1/09, SARS coronavirus (CoV), MERS-CoV, and the more recent outbreaks of Ebola virus in West Africa and Zika virus throughout the Americas.

Importantly, the study of disease emergence has moved from making lists of emerging diseases and the proximate ecological factors responsible for their appearance (~~e.g. see 1~~), to establishing more rigorous quantitative frameworks to explain how new infections become established in populations (~~e.g. 2, 3, 4~~); (e.g. 2, 3, 4). In particular, there is now a large body of literature devoted to understanding the determinants of disease emergence, with a strong focus on emerging viruses. This work has the implicit and commendable goal of trying to reveal the overarching rules of disease emergence which,

in turn, might lead to predictions of what virus might emerge next and where this may occur ~~(5-8)~~(5-8). Gaining this sort of predictive capability would have obvious and wide-ranging benefits. In these approaches, the study of virus emergence is often synonymous with the study of virus evolution, such that the more we understand about the patterns and processes of evolutionary change, the more accurate any emergence prediction is thought to be.

However, accurate predictions of disease emergence must be based on a correct and rigorous understanding of how viruses jump between hosts and adapt to new transmission cycles, including the timescale over which these processes occur. We show here that a more meaningful understanding of virus emergence requires us to shift the focus away from the broader processes of virus evolution and towards the short-term factors that influence the probability of the successful establishment of a virus in a host population. In other words, if the goal is to develop a meaningful predictive model of disease emergence, there may be considerable value in tuning out the background 'noise' of virus evolution rather than building the model around long-term evolutionary processes. More fundamentally, we will argue that a more practical approach to the challenge of virus emergence will involve abandoning prediction in favour of genomic surveillance at the ecological 'fault-lines' of emergence.

2. The Nature of Virus Emergence

The successful emergence of a virus in a new host will often entail a significant adaptive challenge. Indeed, one of the most important observations in disease emergence is that not all viruses that jump species boundaries successfully evolve onward transmission in

the new host. Rather, many such viruses appear as transient 'spill-over' infections that soon die out, even in the absence of infection control. For example, despite repeated spill-over events from birds to humans, H5N1 avian influenza virus has not been able to evolve sustained human-to-human transmission ~~(9),(9)~~. Other viruses have been more successful and resulted in significant outbreaks in new hosts. For example, Ebola virus (EBOV) has caused several localised epidemics that have been largely restricted in their spread by administrative boundaries and border closings ~~(10),(10)~~. Hence, although there is evidence of the active adaptation of EBOV to ~~human populations~~humans during the recent 2013-2016 outbreak in West Africa ~~(11, 12), it is likely that~~(11, 12), the virus clearly possesses the necessary base-line virological traits needed to ensure its onward transmission in the new host. Finally, some viruses have evolved to become endemic human pathogens, involving the generation of well-established and long-standing chains of transmission that do not require repeated spill-over events from an animal reservoir. An obvious case in point is HIV, the agent of AIDS, although a wide variety of human viruses fall into this class. Indeed, it is likely that most endemic virus infections in humans ultimately resulted from cross-species transmission, although in the majority of cases the exact animal reservoir species are unknown or unsampled. For example, although hepaciviruses are being increasingly documented in animal populations, it is likely that the true reservoir species for human hepatitis C virus has yet to be identified (13, 14). These different virus-host associations, from spill-over to endemicity, highlight the two central questions in the evolution of virus emergence: (i) why are only some viruses able to successfully spread in a new host, and (ii) what barriers, both ecological and genetic, prevent active host adaptation from taking place?

There has been a great deal of experimental research in a number of systems directed toward identifying those specific viral genomic mutations responsible for successful host adaptation ~~(13),(15)~~, although as noted above an important limitation is that there are still relatively few cases in which the precise chain of evolutionary events from reservoir to recipient species have been determined (16). As expected, many mutations that promote successful host-adaptation are concerned with aspects of virus-receptor binding ~~(14), although changes in other traits, such as pH, are also of importance (15)~~. However, ~~it is clear that virus genetics alone cannot explain the spectrum of disease emergence types (11, 12, 17)~~, although changes in other traits, such as pH (18) and interactions with host antiviral responses (19, 20), are also of importance. However, virus genetics alone cannot explain why only some emerging viruses are successful. Indeed, even viruses that appear to be well adapted to a specific host (i.e. that seemingly harbour all necessary host-specific mutations) may fail to spread.

An informative example concerns the recent emergence of the A/H3N8 subtype of canine influenza virus (CIV). Although this virus was first recorded in dogs in the USA in the early 2000s, with horses acting as the reservoir host ~~(16),(21)~~, it has failed to become established in the domestic dog population. Instead, CIV is largely confined to dog shelters, where most dogs are infected soon after they arrive ~~(17),(22)~~. CIV clearly possesses all the genetic characteristics necessary to spread in dogs, and its reproductive number in dog shelters is always sufficient (i.e. $R_0 > 1$) to allow its spread within these confined spaces. However, CIV has failed to ignite a wider epidemic in dogs, likely because contact heterogeneity in the domestic dog population is much greater than in dog shelters such that there is an insufficient density of susceptible hosts for the outbreak to take hold ~~(17)~~. ~~This inhibition of virus emergence through a lack of~~

~~susceptibles is likely to be commonplace~~(22). This inhibition of virus emergence through a lack of susceptible hosts is likely to be commonplace. The general lesson to be learned for exercises in prediction is that determining whether a virus can spread in a particular host, for example following cell passage experiments or using animal models, does not mean that it will in the real world unless epidemiological circumstances are permissive. Virus emergence should therefore always be thought of as a combination of successful genetics aligned with permissive ecology ~~(4, 8, 18)~~(4, 8, 16).

3. Is Emergence Predictable?

Predicting emergence has become one of the highest stakes topics in the study of infectious disease. The multi-host dynamics of virus emergence, from ~~donor~~reservoir to recipient hosts, requires us to consider the interplay of host ecology and virus genetics. At the ecological level, emergence risk has been associated with such factors as climate change, population demographics ~~(19)~~, geographic 'hotspots' of former emergence locations (5), and host plasticity (20)~~(23)~~, geographic 'hotspots' of former emergence locations (5), and host plasticity (24). At the virus genetic level, there has been effort to determine those virological factors that correlate with transmissibility within a specific host ~~(7) (see below)~~(7) (see below). Theoretical studies aimed at assessing the predictability of emergence have strived to encompass the complexities of inter- and intra-host evolutionary fitness dynamics ~~(2, 21-23)~~(2, 25-27), where the within-host and between-host fitness landscapes play a central role in determining the probability of emergence ~~(4, 21)~~(4, 25).

A central goal of research in this area has been to reveal the 'rules' that underpin disease emergence, on the implicit assumption that predictive accuracy will follow. A more ambitious scheme was established in 2016 in the guise of the Global Virome Project (GVP). Through a global partnership the GVP aims to identify and characterize 99% of zoonotic viruses with epidemic potential to better predict, prevent and respond to future viral threats (24)(28). To achieve these aims, the GVP will perform large-scale metagenomic surveys of viruses in vertebrate populations. The underlying logic is that knowing what viruses are present in nature provides substantive value in trying to determine what will come next. The GVP is therefore the clarion call for studies in predicting virus emergence.

Surveys of virus biodiversity have already revealed remarkable levels of genomic and phylogenetic diversity (25, 26)(29, 30) and will undoubtedly increase our understanding of the patterns and processes of virus evolution. However, we contend that they are unlikely to be informative in predicting the next pandemic. Most obviously, the total number of viruses on earth – the virosphere – is so large that it can never provide a guided insight into what viruses may eventually emerge in humans. There are an estimated 8.7 million eukaryotic species on earth (27), and it is possible that each could carry in the order of approximately 100 virus species (with more than 200 documented in humans). We can then very roughly estimate that there are in the order of 870 million(31). If we assume that each carries approximately 10 species-specific viruses (more than 200 viruses have been documented in humans), then we can then very roughly estimate that there are in the order of 87 million eukaryotic viruses perhaps distinct enough to be considered different species. Currently, the International Committee on Taxonomy of Viruses (ICTV) recognises 4,404 virus species in all hosts

(both eukaryotes and prokaryotes), which means that 99.999493799949379% of the eukaryotic virosphere remains undiscovered or unclassified (Figure 1). ~~Even if we reduce the number of distinct virus species per host to only 10, 99.9949379% of virus species remain unknown,~~ and these calculations ignore the huge numbers of bacteriophage.

The idea of ~~ana~~ virosphere so expansive is supported by the vast numbers of viruses discovered by recent studies of viral biodiversity that have been stimulated by advances in metagenomics, particularly the use of bulk RNA sequencing (~~25, 26, 28~~)(29, 30, 32). Importantly, these metagenomics studies have considered virus diversity in terrestrial species, whereas previous studies had a strong focus on aquatic environments and DNA bacteriophage (~~29-31~~)(33-35). Most dramatically, a recent metagenomic analysis of nine invertebrate phyla identified 1445 novel RNA viruses, as well as newly defined genera and families (and possibly orders) (~~25~~)(29). Not only does this represent a major increase in our knowledge of virus diversity, but that it came from a survey of only 220 species from a small number of sampling locations in China hints at the true scale of the virosphere.

Although vertebrates, particularly mammals, may carry a smaller number of viruses, the number is still so very large as to make any detailed experimental follow-up of even the vertebrate virosphere impractical, particularly as the rapid nature of RNA virus evolution means that any individual virus species will harbour a wide diversity of ever-changing variants. This impracticality is augmented when one considers our lack of knowledge of whether this vast set of viruses can replicate in human cells, and even this trait will not guarantee that a virus will be able to successfully transmit between hosts. In this context it has been proposed that machine-learning may help in pandemic prediction, for instance by using sequence data to predict which cell receptors a virus might utilize (~~32~~,

~~33~~,(36, 37). However, attaining knowledge of cell receptor compatibility in itself does not enable accurate predictions of emergence, particularly as viruses with a diverse range of receptors are able to infect humans. For example, it has long been known that influenza viruses bind to sialic acid-containing molecules as receptors (34),(38). However, this information has not improved prediction of influenza virus emergence and re-emergence. More generally, machine learning requires very large amounts of data to predict common events, whereas studies of disease emergence necessarily utilize data on *rare* events to predict *rare* events.

Paradoxically, then, the more we sample animal populations, the less frequently virus cross-species transmission to humans seems to occur. For example, when SARS coronavirus (CoV) was revealed to have its origin in bats (35),(39), the total number of known bat viruses was very small so that the likelihood that a bat virus might emerge in humans correspondingly appeared to be relatively high. However, the total number of known bat viruses has increased dramatically with better sampling (6, 36, 37),(6, 40, 41), and bat-to-human zoonotic transmission now appears to be a rare event.

It is also important to recall that the most recent viruses to achieve epidemic spread in humans – Ebola and Zika – were caused by known and well described human pathogens, with the first descriptions of Zika virus going back to the 1940s (38),(42). Yet, our previous knowledge of these pathogens was not indicative of their epidemic potential. It may therefore be the case that the greatest pandemic threat in fact lies in those viruses that re-emerge intermittently ~~and whose onward success depends on the availability of a large, density populated host population~~ in large and dense host populations. We propose a theoretical framework to understand this possibility later.

Previous attempts to predict aspects of virus emergence have met with limited success.

There has been considerable interest in trying to predict ~~those~~the geographic locations where viruses may emerge in the future, based on identifying the locations where such emergence events have occurred in the past: ~~the so-called 'hotspots' of emergence (5); --~~ the so-called 'hotspots' of emergence (5). Although these may well represent localities where sampling virus biodiversity will be profitable, it is difficult to turn such studies into a viable index of predictability. For example, both Mexico and Saudi Arabia appear as 'cold' spots in these emergence maps, with relatively little evidence of past disease emergence. Yet, since these hotspot maps were published, Mexico has witnessed the emergence of H1N1/09 virus (in pig populations), while MERS coronavirus emerged in Saudi Arabia in 2012, with dromedary camels unexpectedly acting as the reservoir host ~~(39);(43)~~.

It has also been suggested that wildlife host species richness is an important predictor of disease emergence ~~(5);(5)~~. Conversely, however, biodiversity has also been linked to a decrease in disease risk through the 'dilution effect' ~~(40-44);(44-48)~~. This was first developed as a framework to infer the dynamics of tick-borne Lyme disease and describes the association between increasing species richness and reduced disease risk, particularly when the most competent hosts were dominant in the community and alternative hosts negatively influenced the dominant hosts as reservoirs ~~(40, 45);(44, 49)~~. Although still debated, the dilution effect highlights the central role of host biodiversity and ecology in shaping the epidemiology of disease-causing pathogens. Inevitably, habitat destruction and ecosystem disturbance due to changes in land use will contribute to the loss of biodiversity. The broader consequences of such losses for emerging human pathogens are unknown and clearly merit further investigation (Figure 2).

Although cross-species virus transmission sits at the heart of virus emergence, phylogenetic studies of the frequency with which different virus families are able to jump species boundaries also offer little predictive power as all exhibit a strong tendency to jump hosts ~~(46)~~. Indeed, it now appears that the evolutionary history of most virus families comprises a complex mix of cross-species transmission and virus-host co-divergence, and that trying to disentangle the respective contributions of each process will be challenging ~~(46)~~(50). In addition, the greater diversity of hosts and their viruses sampled, the more cases of species jumping we are likely to document ~~(46)~~(50). Importantly, these phylogenetic studies also demonstrate that virus-host associations, including cross-transmission, may extend over many millions of years and not only in the recent past as is assumed in studies of virus emergence. ~~As a case in point, evolutionary relationships within the Narna–Levi group of RNA viruses are compatible with virus-host co-divergence since the α -proteobacteria became endosymbionts (25).~~ As a case in point, it is possible that the Narna–Levi group of RNA viruses have co-diverged with their hosts since the α -proteobacteria became endosymbionts (29).

While other comparative analyses have revealed those virological factors that increase the transmissibility of emerging viruses in humans ~~(7)~~(7), these analyses also likely offer little predictive power. These studies suggest that viruses with low host mortality, that establish chronic infections, that are non-segmented, that do not possess an envelope, and that are not transmitted by vectors have greater 'emergibility' in humans ~~(7)~~(7). Nonetheless, many viruses still fall into this class and a number of these traits are not measurable until the virus has already established itself in a new host, diminishing the predictive utility of such 'viral traits' analyses.

Given these uncertainties, and the fact that elements of the evolutionary ~~processes~~process that underpins emergence are inherently unpredictable, we suggest that there is no simple algorithm that will enable an accurate prediction of what viruses might emerge in the future. ~~Accordingly, we suggest that~~Hence, it is necessary to lower our expectations about disease emergence as a predictive science. In particular, although metagenomics undoubtedly has major implications for our understanding of virus evolution, it also likely undermines biodiversity-based attempts to predict the virus source of the next major disease pandemic ~~(6),(6)~~. There are clearly so many viruses in nature that trying to determine which will ultimately appear in a new host from diversity sampling alone is almost certainly futile.

Predictions also sit uneasily with most aspects of evolutionary biology. Even relatively simple traits like virulence, which have generated considerable evolutionary theory, have proven difficult to predict because of myriad unknown forces that shape their evolutionary trajectory ~~(47),(51)~~. Although there has been some success in using phylogenetic approaches to predict the short-term evolution of human influenza virus ~~(48),(52)~~, the nature of the central selective processes shaping virus evolution (i.e. antigenic drift) is well known and to some extent quantifiable over the timescale studied. This is demonstrably not the case when considering unknown emerging viruses.

4. The conflation of epidemiological and evolutionary timescales

At face value it seems obvious that evolutionary ideas and analyses will help predict the emergence threat posed by different viruses. However, a major limitation is that evolutionary processes, particularly those reliant on phylogenetic or other comparative

analyses, often occur on a markedly different timescale than the epidemiological processes relevant to pandemic prediction. Indeed, one of the most important conclusions of recent work in the study of RNA virus evolution is that the timescale over which these viruses have evolved, including cross-species transmission events, is likely far longer than previously imagined. This realisation comes from both phylogenetic studies of virus biodiversity and branching patterns ~~(46, 49)~~, (50, 53), particularly the match between parts of the virus and host trees, and the analyses of endogenous virus elements that act as genomic fossils ~~(50)~~, (54). Hence, it is likely that many of the viral families that infect vertebrates have done so for many millions of years, and have experienced continual cross-species transmission since this time.

Although central to understanding evolutionary processes, these timescales are irrelevant for predicting the next pandemic within an epidemiological timescale (i.e. 1-10 years). The same caveat applies to studies that have used the taxonomic span covered by viral families as a way of determining which have the greatest propensity to jump hosts ~~(6)~~, (6). These taxonomic ranges may have taken millions of years to generate and not the scale of years necessary for effective pandemic prediction. Evolutionary and epidemiological timescales should not simply be assumed to be equivalent. Although phylogenies can be used to accurately describe both macro- and micro-evolution, and superficially appear similar, the trees at these two scales are produced by markedly different evolutionary processes (Figure 3). As it is clear that the pace of human ecological (anthropogenic) change generally occurs more rapidly than successful virus host-jumping adaptations depicted in a phylogenetic tree, from a public health point of view we would do better to monitor ongoing environmental disturbance by humans than quantify long-term aspects of virus evolution.

An informative example of this fundamental disconnect between evolutionary and epidemiological timescales is provided by the hepadnaviruses, which include human hepatitis B virus (HBV). There is strong evidence for hepadnavirus-host co-divergence stretching back for effectively the entire time-span of vertebrate evolution (49), (53), (55). Cross-species transmission has occurred on this background of co-divergence, with a recent analysis revealing ~13 host jumps over an evolutionary period of approximately 400 million years (46). ~~Although our sample of hepadnaviruses is inevitably small,~~ (50). Although our sample of hepadnaviruses is inevitably small, with new hepadnaviruses recently identified in fish (55), and more cases of cross-species transmission will assuredly be found, this very roughly equates to a successful cross species transmission event every 30 million years. Even if the rate of host jumping is 10,000 times more frequent, occurring once every 3,000 years, this is still far too broad brush a timescale to provide any meaningful predictive value for the study of human disease emergence. A similar story can be told for the influenza viruses. Although these are exemplars of cross-species transmission (51), ~~which occurs frequently in the *Orthomyxoviridae*~~ (46), (56), which occurs frequently in the *Orthomyxoviridae* (50), it is still problematic to make these predictions over the timescale of human observation. For example, the emergence of H3N8 equine influenza virus from an avian host took place in the early 1960s. Although this virus is clearly adapted for mammalian respiratory transmission, there is no evidence that it has transmitted to humans during the last 50 years.

Those cases in which viruses have been deliberately released as biological controls also highlight the disconnect between evolutionary and epidemiological timescales. These natural experiments proceed over epidemiological timescales which in many ways parallel the natural emergence and spread of a novel virus in a new host. Most notably,

both myxomavirus (a poxvirus) and rabbit haemorrhagic disease virus (a calicivirus) have been successfully released as biological controls into populations of European rabbits in Australia and Europe, in the 1950s and 1990s, respectively ~~(52). What is particularly striking is that to date there are no~~(57). What is particularly striking is that to date there are no strongly supported cases of these viruses jumping into other (i.e. non-lagomorph) species over the timescale of release, even though both ~~these virus families appear to experience very frequent host jumping over long evolutionary timescales~~ (46)-virus families appear to experience very frequent host-jumping over long evolutionary timescales (50). Hence, the observation that poxviruses can frequently jump species boundaries over evolutionary timescales provides no assistance in predicting what happens on the shorter ~~timescales~~timescale that govern epidemics.

5. A population-genetic framework to understand virus emergence

The study of virus emergence represents a synthesis of two different types of scientific enquiry: virology, which aims to determine, usually experimentally, the mutations that enable a virus to infect a new host, and epidemiology, which primarily seeks to identify the ecological factors responsible for viruses crossing the species boundary and spreading in a new host.

We ~~believe~~contend that ~~both~~ these approaches can be synthesised within a single population genetic framework. Specifically, the cross-species transmission and emergence of a virus in a new host ~~might~~can be ~~explained~~envisioned as a simple form of the adaptive process, ~~wherein~~ in which the subject under consideration is the acquisition of mutations that facilitate replication and transmission and hence increase viral fitness.

Although they may be of myriad form, the ecological factors that dictate whether such an emergence event will be successful are directly analogous to the random sampling effects that necessarily impact the spread of any new allele in a population, increasing the likelihood of genetic drift that will in turn result in stochastic loss. For example, the extensive contact heterogeneity (i.e. lack of susceptible hosts) that prevents CIV spreading in the domestic dog population is equivalent to the fate of an advantageous allele in a small host population. That is, although the virus (mutation) may be host adapted (advantageous), it will not spread far because the host population is so small/sparse that genetic drift dominates substitution dynamics (and even strongly advantageous alleles may be lost rather than fixed in small populations).

We suggest that this new population genetic view of the process of cross-species transmission and emergence can be achieved by making the move away from thinking about viruses spreading horizontally through a population (by host-to-host transmission), which is the realm of epidemiology, and towards thinking about virus alleles/genes being inherited vertically, which is the domain of population genetics. A well-understood framework is that successful cross-species transmission requires three steps: (i) encounter a new host species, (ii) infection of that new host, and (iii) propagation in ~~new host population (53)~~. Genetic adaptation to the new host population (58). Adaptation to the new host species may often represent a major challenge, as mutations that are beneficial in this host also likely decrease fitness in the ~~donor host species~~. This opposing selection between donor and recipient hosts shapes the adaptive landscape of viral emergence (21). reservoir host species. This opposing selection between reservoir and recipient hosts shapes the adaptive landscape of viral emergence (25). For example, as the gradient of the adaptive landscape increases, genetic variants are subjected to stronger opposing

selection between the ~~donor~~reservoir and the recipient hosts. Models of this adaptive process therefore offer an indication of which part of parameter space the host adaptation of a novel virus might be possible (~~2, 4, 21, 23~~); (2, 4, 25, 27).

Importantly, however, this adaptive process must also occur within the background of random sampling. Because of a lack of host contacts, or descendants, genetic drift will reduce sampling of the fittest virus and decrease the probability of emergence. Hence, as the host population increases in size, the probability that a virus will be sampled increases (Figure 4); although it is likely that additional factors, including prior immunity and population age structure, will also impact the probability of virus sampling. A simple lesson from this new realisation approach is that host population size and density – which can be thought of as comprising the 'effective susceptible population size' for an emerging virus – will have a major impact on whether a new virus will successfully spread in a population, irrespective of the fitness of a particular mutation (i.e. whether a virus contains all the mutations necessary to adapt it to a new host). Consequently, if the fitness of a virus and the ~~host~~effective susceptible population size were known, or even measurable, it would be possible to make bounded estimates of how likely a successful emergence event might be.

The importance of genetic drift can also be seen in the transmission bottlenecks that will routinely occur as a virus moves between hosts (~~54~~); which likely puts a brake on host adaptation (~~55–58~~); (59); which likely put a brake on host adaptation (~~59–63~~). Even if a specific variant is favoured within an individual host, but does not increase sufficiently in frequency (i.e. such that is still found at sub-consensus levels), then a severe population bottleneck may result in its loss. Clearly, the more severe the population bottleneck, the less natural selection will be able to optimise viral fitness at the epidemiological scale.

Formatted: Font color: Auto

Despite the insights provided by this population genetic framework, the vastness of the unknown virosphere means that all such theories of emergence are probably of more use in predicting the population impact of emergence rather than predicting what might emerge next and in what location.

6. Sampling the human-animal interface: fault-lines of disease emergence

If the accurate prediction of virus emergence is impractical, or even unattainable, what can be done to help prevent the emergence of major epidemic or pandemic disease in humans? Cross-species transmission events highlight the public health threat of wildlife trade and consumption, and heightened contact between animal hosts and humans has likely facilitated cross-species transmission and provided increased opportunity for transmission events. It is therefore critical to direct our attention to the animal-human interface (humans, livestock trade and consumption, wildlife, environment) as this can be thought of as the 'fault-line' at which most disease emergence events occur.

We therefore propose that a more effective practical strategy for managing emerging and re-emerging epidemic or pandemic disease is the targeted surveillance of viromes at the human-animal interface. The vast biodiversity of viruses in the animal world makes their analysis prior to any emergence in humans a Sisyphean exercise. Rather, humans are the best sentinels: a virus discovered in humans very obviously can replicate in that host, which will not be the case for myriad viruses identified through biodiversity surveys of other animal taxa.

We therefore urge regular genomic surveillance at the fault-line of disease emergence that captures this human-animal interface (Figure 4). Examples of this interface that could be sampled are those associated with (i) major changes in land-use, particularly human encroachment into forest areas during deforestation; (ii) occupational exposure to live animal markets; and (iii) changes in human demographics, behaviour and political instability that result in population mobility and displacement. To take one specific example, the hunting and butchering of wild animals, and the meat trade that flows from it, is common practice among many countries. ~~This activity must represent a conduit for cross-species pathogen transmission, and is likely responsible for the transfer of simian retroviruses from infected nonhuman primates to humans (59). Virological surveillance of those working in the bushmeat trade therefore appears a necessary measure. Importantly, accelerating environmental and anthropogenic changes are expanding the human-animal interface (59);~~This activity must represent a conduit for cross-species pathogen transmission, and is likely responsible for the transfer of simian retroviruses from infected non-human primates to humans (64). Virological surveillance of those working in the bushmeat trade therefore appears a necessary measure. Importantly, accelerating environmental and anthropogenic changes are expanding the human-animal interface (59), and the rapid movement of humans and livestock, as well as agricultural produce, highlights the importance of effective surveillance.

This virome surveillance should be ongoing and performed simultaneously on multiple human populations globally. ~~While, with existing serological data perhaps helping to determine which geographical locations harbour human populations most frequently exposed to animal viruses and hence where virome surveillance will be most informative.~~In addition, while metagenomics is hugely powerful in characterising the viromes of

individual organisms, including the discovery of new species, it requires active infection (replication) and that samples be taken from tissues that contain the virus. For this reason meta-serological surveys will also be of importance as they enable the identification of infections that have occurred in the recent past.

7. Conclusions

Predicting virus emergence has risen to become a key goal of the study of infectious disease. The study of virus evolution has revealed much about the nature of virus emergence and its history over evolutionary timescales. However, due to the fundamental differences between evolutionary and epidemiological timescales, a focus on virus evolution may in fact be a distraction when it comes to predicting the next virus pandemic. Similarly, while virological features that increase the likelihood of virus emergence can be identified, these features cannot be treated as hard and fast rules determining which viruses will ~~in fact~~ successfully emerge. Further, many of these features are only capable of being observed *after* emergence occurs, such that they are likely to be of little predictive power. In partial response to these problems, we suggest that the field may be advanced by utilizing a population genetic framework that melds genetic and ecological studies of virus emergence, and which highlights how the effective susceptible population size of a new host plays a major role in dictating the chance of successful emergence. In this manner we identify the possibility of a meaningful theoretical framework for the study of emergence that is grounded in evolutionary theory, but that tunes out the 'noise' of virus macroevolution.

Despite such a framework, the inconvenient truth for all those working in the realm of disease emergence is that the vastness of the unknown virosphere and the diverse range of viruses that have achieved endemic transmission in humans means that any attempt to predict what virus may emerge next will face substantial, and likely crippling, difficulties. In light of this we suggest it may be of more benefit to public health to target, via surveillance, the fault-line of disease emergence that is the human-animal interface, particular those shaped by ecological disturbance. Once a virus is identified as being of interest in this manner, other analyses may be able to assess its impact and pandemic potential. Such a shift in focus, away from being able to make predictions of emergence based on fundamental rules and toward the better assessment of emergence impact, is ~~likely~~ both more achievable and more likely to provide positive public health outcomes.

Competing interests. We declare we have no competing interests.

Author's contributions. JLG and ECH jointly conceived and wrote the paper. Both authors gave final approval for publication.

Funding. ECH is funded by an Australia Fellowship from the National Health and Medical Research Council, Australia (GNT1037231).

Figure Legends

Figure 1. Illustration of the relative size of the potentially unknown virosphere. The estimates shown are based on the assumption that approximately ~~10010~~ viruses might be capable of infecting each of the estimated 8.7 million eukaryotic species on earth ~~(27)~~(27). Currently, the ICTV recognises 4404 virus species in both eukaryotes and prokaryotes (although many others are awaiting classification), which means that 99.999493799949379% of the virosphere remains undiscovered or unclassified.

Figure 2. Possible effects of host biodiversity on the probability of viral emergence. The red arrows at the bottom depict instances of increased emergence risk. Wildlife host species richness has been proposed as an important predictor of disease emergence. Likewise, host populations of low biodiversity might harbour fewer viruses and a decreased risk of emergence. Conversely, high host biodiversity has also been linked to a decrease in disease risk through the 'dilution effect'.

Figure 3. Phylogenetic analyses of a virus family, seemingly showing many instances of cross-species transmission over evolutionary timescales (i.e. virus macroevolution). Critically, however, the adaptive processes (i.e. mutation and selection) that lead to virus 'spill-overs' and possible emergence in a new host are more informative when considering a shorter, microevolutionary timescale.

Figure 4. Exemplifying the human-animal interface as the fault-line of disease emergence. Following a cross-species transmission event, a virus might cause a dead end

'spill-over' or it might be genetically adapted to be transmissible between members of the new host species. ~~The~~Even for emergent viruses of high fitness the probability of emergence and the size of the outbreak relies on a large and dense host population, ~~even for emergent viruses of high fitness,~~as well as a variety of other ecological factors that can be thought of as comprising the 'effective susceptible population size' (x-axis, right hand panel).

References

1. — Morse SS. 1995 Factors in the emergence of infectious diseases. *Emerg. Infect. Dis.* **1**, 1–7–15. (doi:10.3201/eid0101.950102)
2. — Russell CA, Fonville JM, Brown AEX, Burke DF, Smith DL, James SL, Herfst S, van Boheemen S, Linster M, Schrauwen EJ, et al. 2012 The potential for respiratory droplet transmissible A/H5N1 influenza virus to evolve in a mammalian host. *Science* **336**, 6088–1541–1547. (doi:10.1126/science.1222526)
3. — Blumberg S, Lloyd-Smith JO. 2013 Inference of R0 and transmission heterogeneity from the size distribution of stuttering chains. *PLoS Comput. Biol.* **9**, 5 e1002993. (doi:10.1371/journal.pcbi.1002993)
4. — Park M, Loverdo C, Schreiber SJ, Lloyd-Smith JO. 2013 Multiple scales of selection influence the evolutionary emergence of novel pathogens. *Philos. Trans. R. Soc. Lond. B. Biol. Sci.* **368**, 1614 (doi:10.1098/rstb.2012.0333)
5. — Jones KE, Patel NG, Levy MA, Storeygard A, Balk D, Gittleman JL, Daszak P. 2008 Global trends in emerging infectious diseases. *Nature* **451**, 990–993. (doi:10.1038/nature06536)
6. — Olival KJ, Hosseini PR, Zambrana-Torrel C, Ross N, Bogich TL, Daszak P. 2017 Host and viral traits predict zoonotic spillover from mammals. *Nature* **546**, 646–650. (doi:10.1038/nature22975)
7. — Geoghegan JL, Senior AM, Di Giallonardo F, Holmes EC. 2016 Virological factors that increase the transmissibility of emerging human viruses. *Proc. Natl. Acad. Sci. U.S.A.* **113**, 4170–4175. (doi:10.1073/pnas.1521582113)
8. — Plowright RK, Parrish CR, McCallum H, Hudson PJ, Ko AI, Graham AL, Lloyd-Smith JO. 2017 Pathways to zoonotic spillover. *Nat. Rev. Micro.* In press. (doi:10.1038/nrmicro.2017.45)

9. — Lam TT, Zhou B, Wang J, Chai Y, Shen Y, Chen X, Ma C, Hong W, Chen Y, Zhang Y, et al. 2015 Dissemination, divergence and establishment of H7N9 influenza viruses in China. *Nature* **522**, 7554–102–105. (doi:10.1038/nature14348)
10. — Dudas G, Carvalho LM, Bedford T, Tatem AJ, Baele G, Faria NR, Park DJ, Ladner JT, Arias A, Asogun D, et al. 2017 Virus genomes reveal factors that spread and sustained the Ebola epidemic. *Nature* **544**, 7650–309–315. (doi:10.1038/nature22040)
11. — Urbanowicz Richard A, McClure C P, Sakuntabhai A, Sall Amadou A, Kobinger G, Müller Marcel A, Holmes Edward C, Rey Félix A, Simon Loriere E, Ball Jonathan K. 2016 Human adaptation of Ebola virus during the West African outbreak. *Cell* **167**, 4–1079–1087.e1075. (doi:10.1016/j.cell.2016.10.013)
12. — Diehl WE, Lin AE, Grubaugh ND, Carvalho LM, Kim K, Kyawe PP, McCauley SM, Donnard E, Kucukural A, McDonel P, et al. 2016 Ebola virus glycoprotein with increased infectivity dominated the 2013–2016 epidemic. *Cell* **167**, 4–1088–1098.e1086. (doi:10.1016/j.cell.2016.10.014)
13. — Herfst S, Schrauwen EJ, Linster M, Chutinimitkul S, de Wit E, Munster VJ, Sorrell EM, Bestebroer TM, Burke DF, Smith DJ, et al. 2012 Airborne transmission of influenza A/H5N1 virus between ferrets. *Science* **336**, 6088–1534–1541. (doi:10.1126/science.1213362)
14. — Taubenberger JK, Kash JC. 2010 Influenza virus evolution, host adaptation and pandemic formation. *Cell Host Microbe* **7**, 6–440–451. (doi:10.1016/j.chom.2010.05.009)
15. — Zaraket H, Bridges OA, Russell CJ. 2013 The pH of activation of the hemagglutinin protein regulates H5N1 influenza virus replication and pathogenesis in mice. *J. Virol.* **87**, 9–4826–4834. (doi:10.1128/jvi.03110-12)
16. — Crawford PC, Dubovi EJ, Castleman WL, Stephenson I, Gibbs EP, Chen L, Smith C, Hill RC, Ferro P, Pompey J, et al. 2005 Transmission of equine influenza virus to dogs. *Science* **310**, 5747–482–485. (doi:10.1126/science.1117950)
17. — Dalziel BD, Huang K, Geoghegan JL, Arinaminpathy N, Dubovi EJ, Grenfell BT, Ellner SP, Holmes EC, Parrish CR. 2014 Contact heterogeneity, rather than transmission

efficiency, limits the emergence and spread of canine influenza virus. *PLoS Pathog.* **10**, 10 e1004455. (doi:10.1371/journal.ppat.1004455)

18.— Parrish CR, Holmes EC, Morens DM, Park EC, Burke DS, Calisher CH, Laughlin CA, Saif LJ, Daszak P. 2008 Cross-species virus transmission and the emergence of new epidemic diseases. *Microbiol. Mol. Biol. Rev.* **72**, 3 457–470. (doi:10.1128/mmb.00004-08)

19.— Stephen SM. 1995 Factors in the emergence of infectious diseases. *Emerg. Infect. Dis.* **1**, 1 7. (doi:10.3201/eid0101.950102)

20.— Johnson CK, Hitchens PL, Evans TS, Goldstein T, Thomas K, Clements A, Joly DO, Wolfe ND, Daszak P, Karesh WB, et al. 2015 Spillover and pandemic properties of zoonotic viruses with high host plasticity. *Sci. Rep.* **5**, (doi:10.1038/srep14830)

21.— Geoghegan JL, Senior AM, Holmes EC. 2016 Pathogen population bottlenecks and adaptive landscapes: overcoming the barriers to disease emergence. *Proc. R. Soc. Lond. B. Biol.* **283**, 1837 (doi:10.1098/rspb.2016.0727)

22.— Lipsitch M, Barclay W, Raman R, Russell CJ, Belser JA, Cobey S, Kassin PM, Lloyd-Smith JO, Maurer-Stroh S, Riley S, et al. 2016 Viral factors in influenza pandemic risk assessment. *eLife* **5**, (doi:10.7554/eLife.18491)

23.— Fonville JM. 2015 Expected effect of deleterious mutations on within host adaptation of pathogens. *J. Virol.* **89**, 18 9242–9251. (doi:10.1128/jvi.00832-15)

24.— Daszak P, Carroll D, Wolfe N, Mazet J. 2016 The global virome project. *Int. J. Infect. Dis.* **53**, 36. (doi:10.1016/j.ijid.2016.11.097)

25.— Shi M, Lin X-D, Tian J-H, Chen L-J, Chen X, Li C-X, Qin X-C, Li J, Cao J-P, Eden J-S, et al. 2016 Redefining the invertebrate RNA virosphere. *Nature* **540**, 7634–539–543. (doi:10.1038/nature20167)

26.— Li C-X, Shi M, Tian J-H, Lin X-D, Kang Y-J, Chen L-J, Qin X-C, Xu J, Holmes EC, Zhang Y-Z. 2015 Unprecedented genomic diversity of RNA viruses in arthropods reveals the ancestry of negative sense RNA viruses. *eLife* **4**, e05378. (doi:10.7554/eLife.05378)

27. — Mora C, Tittensor DP, Adl S, Simpson AGB, Worm B. 2011 How many species are there on Earth and in the ocean? *PLoS Biol.* **9**, 8 e1001127. (doi:10.1371/journal.pbio.1001127)
28. — Webster CL, Waldron FM, Robertson S, Crowson D, Ferrari G, Quintana JF, Brouqui JM, Bayne EH, Longdon B, Buck AH, et al. 2015 The discovery, distribution, and evolution of viruses associated with *Drosophila melanogaster*. *PLoS Biol.* **13**, 7 e1002210. (doi:10.1371/journal.pbio.1002210)
29. — Culley AI, Lang AS, Suttle CA. 2006 Metagenomic analysis of coastal RNA virus communities. *Science* **312**, 5781–1795–1798. (doi:10.1126/science.1127404)
30. — Desnues C, Rodriguez Brito B, Rayhawk S, Kelley S, Tran T, Haynes M, Liu H, Furlan M, Wegley L, Chau B, et al. 2008 Biodiversity and biogeography of phages in modern stromatolites and thrombolites. *Nature* **452**, 7185–340–343. (doi:10.1038/nature06735)
31. — Paez-Espino D, Eloe-Fadrosh EA, Pavlopoulos GA, Thomas AD, Huntemann M, Mikhailova N, Rubin E, Ivanova NN, Kyrpides NC. 2016 Uncovering Earth's virome. *Nature* **536**, 7617–425–430. (doi:10.1038/nature19094)
32. — Yao Y, Li X, Liao B, Huang L, He P, Wang F, Yang J, Sun H, Zhao Y, Yang J. 2017 Predicting influenza antigenicity from Hemagglutinin sequence data based on a joint random forest method. *Sci. Rep.* **7**, 1–1545. (doi:10.1038/s41598-017-01699-z)
33. — Woolhouse MEJ, Ashworth JL. 2017 Can we identify viruses with pandemic potential? *The Biochemist* **39**, 3–8–11. biochemistry.org
34. — García-Sastre A. 2010 Influenza virus receptor specificity: disease and transmission. *Am. J. Pathol.* **176**, 4–1584–1585. (doi:10.2353/ajpath.2010.100066)
35. — Li W, Shi Z, Yu M, Ren W, Smith C, Epstein JH, Wang H, Crameri G, Hu Z, Zhang H, et al. 2005 Bats are natural reservoirs of SARS-like coronaviruses. *Science* **310**, 5748–676–679. (doi:10.1126/science.1118391)

- 36.— Anthony SJ, Johnson CK, Greig DJ, Kramer S, Che X, Wells H, Hicks AL, Joly DO, Wolfe ND, Daszak P, et al. 2017 Global patterns in coronavirus diversity. *Virus Evol.* **3**, 1 vex012–vex012. (doi:10.1093/ve/vex012)
- 37.— Luis AD, Hayman DT, O'Shea TJ, Cryan PM, Gilbert AT, Pulliam JR, Mills JN, Timonin ME, Willis CK, Cunningham AA, et al. 2013 A comparison of bats and rodents as reservoirs of zoonotic viruses: are bats special? *Proc. Biol. Sci.* **280**, 1756–20122753. (doi:10.1098/rspb.2012.2753)
- 38.— Dick GW, Kitchen SF, Haddow AJ. 1952 Zika virus. I. Isolations and serological specificity. *Trans. R. Soc. Trop. Med. Hyg.* **46**, 5–509–520. (doi:doi.org/10.1016/0035-9203(52)90042-4)
- 39.— Milne-Price S, Miazgowiec KL, Munster VJ. 2014 The emergence of the Middle East Respiratory Syndrome coronavirus (MERS-CoV). *Pathog. Dis.* **71**, 2–119–134. (doi:10.1111/2049-632X.12166)
- 40.— Schmidt KA, Ostfeld RS. 2001 Biodiversity and the dilution effect in disease ecology. *Ecology* **82**, 3–609–619. (doi:10.1890/0012-9658(2001)082[0609:BATDEI]2.0.CO;2)
- 41.— Johnson PTJ, Preston DL, Hoverman JT, Richgels KLD. 2013 Biodiversity decreases disease through predictable changes in host community competence. *Nature* **494**, 7436–230–233. (doi:10.1038/nature11883)
- 42.— Allan BF, Langerhans RB, Ryberg WA, Landesman WJ, Griffin NW, Katz RS, Oberle BJ, Schutzenhofer MR, Smyth KN, de St Maurice A, et al. 2009 Ecological correlates of risk and incidence of West Nile virus in the United States. *Oecologia* **158**, 4–699–708. (doi:10.1007/s00442-008-1169-9)
- 43.— Clay CA, Lehmer EM, Jeor SS, Dearing MD. 2009 Sin Nombre virus and rodent species diversity: a test of the dilution and amplification hypotheses. *PLOS ONE* **4**, 7–e6467. (doi:10.1371/journal.pone.0006467)
- 44.— Haas SE, Hooten MB, Rizzo DM, Meentemeyer RK. 2011 Forest species diversity reduces disease risk in a generalist plant pathogen invasion. *Ecol. Lett.* **14**, 11–1108–1116. (doi:10.1111/j.1461-0248.2011.01679.x)

- 45.— LoGiudice K, Ostfeld RS, Schmidt KA, Keesing F. 2003 The ecology of infectious disease: effects of host diversity and community composition on Lyme disease risk. *Proc. Natl. Acad. Sci. U.S.A.* **100**, 2 567–571. (doi:10.1073/pnas.0233733100)
- 46.— Geoghegan JL, Duchêne S, Holmes EC. 2017 Comparative analysis estimates the relative frequencies of co-divergence and cross-species transmission within viral families. *PLoS Pathog.* **13**, 2 e1006215. (doi:10.1371/journal.ppat.1006215)
- 47.— Read AF. 1994 The evolution of virulence. *Trends Microbiol.* **2**, 3 73–76. (doi:10.1016/0966-842X(94)90537-1)
- 48.— Luksza M, Lassig M. 2014 A predictive fitness model for influenza. *Nature* **507**, 7490 57–61. (doi:10.1038/nature13087)
- 49.— Dill JA, Camus AC, Leary JH, Di Giallonardo F, Holmes EC, Ng TF. 2016 Distinct viral lineages from fish and amphibians reveal the complex evolutionary history of hepadnaviruses. *J. Virol.* **90**, 17 7920–7933. (doi:10.1128/jvi.00832-16)
- 50.— Katzourakis A, Gifford RJ. 2010 Endogenous viral elements in animal genomes. *PLoS Genet.* **6**, 11 e1001191. (doi:10.1371/journal.pgen.1001191)
- 51.— Horimoto T, Kawaoka Y. 2005 Influenza: lessons from past pandemics, warnings from current incidents. *Nat. Rev. Microbiol.* **3**, 8 591–600. (doi:10.1038/nrmicro1208)
- 52.— Di Giallonardo F, Holmes EC. 2015 Exploring host-pathogen interactions through biological control. *PLoS Pathog.* **11**, 6 e1004865. (doi:10.1371/journal.ppat.1004865)
- 53.— Moury B, Fabre F, Hébrard E, Froissart R. 2017 Determinants of host species range in plant viruses. *J. Gen. Virol.* **98**, 4 862–873. (doi:doi:10.1099/jgv.0.000742)
- 54.— Leonard AS, Weissman D, Greenbaum B, Ghedin E, Koelle K. 2017 Transmission bottleneck size estimation from pathogen deep sequencing data, with an application to human influenza A virus. *J. Virol.* **91**, 14 e00171–00117. (doi:10.1128/JVI.00171-17)
- 55.— Zwart MP, Elena SF. 2015 Matters of size: genetic bottlenecks in virus infection and their potential impact on evolution. *Annu Rev Virol* **2**, 1 161–179. (doi:doi:10.1146/annurev-virology-100114-055135)

56. — Grubaugh ND, Fauver JR, Ruckert C, Weger Lucarelli J, Garcia Luna S, Murrieta RA, Gendernalik A, Smith DR, Brackney DE, Ebel GD. 2017 Mosquitoes transmit unique West Nile virus populations during each feeding episode. *Cell Rep.* **19**, 4 709–718. (doi:10.1016/j.celrep.2017.03.076)

57. — Stack JC, Murcia PR, Grenfell BT, Wood JL, Holmes EC. 2013 Inferring the inter-host transmission of influenza A virus using patterns of intra-host genetic variation. *Proc Biol Sci* **280**, 1750–20122173. (doi:10.1098/rspb.2012.2173)

58. — Moncla LH, Zhong G, Nelson CW, Dinis JM, Mutschler J, Hughes AL, Watanabe T, Kawaoka Y, Friedrich TC. 2016 Selective bottlenecks shape evolutionary pathways taken during mammalian adaptation of a 1918-like avian influenza virus. *Cell Host Microbe* **19**, 2 169–180. (doi:10.1016/j.chom.2016.01.011)

59. — Kelly TR, Karesh WB, Johnson CK, Gilardi KVK, Anthony SJ, Goldstein T, Olson SH, Machalaba C, Mazet JAK. 2017 One Health proof of concept: Bringing a transdisciplinary approach to surveillance for zoonotic viruses at the human-wild animal interface. *Prev. Vet. Med.* **137**, 112–118. (doi:10.1016/j.prevetmed.2016.11.023)

1. — Morse SS. 1995 Factors in the emergence of infectious diseases. *Emerg. Infect. Dis.* **1**, 7–15. (doi:10.3201/eid0101.950102)

2. — Russell CA, Fonville JM, Brown AEX, Burke DF, Smith DL, James SL, Herfst S, van Boheemen S, Linster M, Schrauwen EJ, et al. 2012 The potential for respiratory droplet transmissible A/H5N1 influenza virus to evolve in a mammalian host. *Science* **336**, 1541–1547. (doi:10.1126/science.1222526)

3. — Blumberg S, Lloyd-Smith JO. 2013 Inference of R_0 and transmission heterogeneity from the size distribution of stuttering chains. *PLoS Comput. Biol.* **9**, e1002993. (doi:10.1371/journal.pcbi.1002993)

4. — Park M, Loverdo C, Schreiber SJ, Lloyd-Smith JO. 2013 Multiple scales of selection influence the evolutionary emergence of novel pathogens. *Philos. Trans. R. Soc. Lond. B. Biol. Sci.* **368**, 20120333. (doi:10.1098/rstb.2012.0333)

5. — Jones KE, Patel NG, Levy MA, Storeygard A, Balk D, Gittleman JL, Daszak P. 2008 Global trends in emerging infectious diseases. *Nature* **451**, 990–993. (doi:10.1038/nature06536)

6. Olival KJ, Hosseini PR, Zambrana-Torrel C, Ross N, Bogich TL, Daszak P. 2017 Host and viral traits predict zoonotic spillover from mammals. *Nature* **546**, 7660 646-650. (doi:10.1038/nature22975)
7. Geoghegan JL, Senior AM, Di Giallonardo F, Holmes EC. 2016 Virological factors that increase the transmissibility of emerging human viruses. *Proc. Natl. Acad. Sci. U.S.A.* **113**, 4170-4175. (doi:10.1073/pnas.1521582113)
8. Plowright RK, Parrish CR, McCallum H, Hudson PJ, Ko AI, Graham AL, Lloyd-Smith JO. 2017 Pathways to zoonotic spillover. *Nat. Rev. Micro.* **15**, 502-510. (doi:10.1038/nrmicro.2017.45)
9. Lam TT, Zhou B, Wang J, Chai Y, Shen Y, Chen X, Ma C, Hong W, Chen Y, Zhang Y, et al. 2015 Dissemination, divergence and establishment of H7N9 influenza viruses in China. *Nature* **522**, 102-105. (doi:10.1038/nature14348)
10. Dudas G, Carvalho LM, Bedford T, Tatem AJ, Baele G, Faria NR, Park DJ, Ladner JT, Arias A, Asogun D, et al. 2017 Virus genomes reveal factors that spread and sustained the Ebola epidemic. *Nature* **544**, 309-315. (doi:10.1038/nature22040)
11. Urbanowicz RA, McClure CP, Sakuntabhai A, Sall AA, Kobinger G, Müller MA, Holmes EC, Rey FA, Simon-Loriere E, Ball JK. 2016 Human adaptation of Ebola virus during the West African outbreak. *Cell* **167**, 1079-1087. (doi:10.1016/j.cell.2016.10.013)
12. Diehl WE, Lin AE, Grubaugh ND, Carvalho LM, Kim K, Kyawe PP, McCauley SM, Donnard E, Kucukural A, McDonel P, et al. 2016 Ebola virus glycoprotein with increased infectivity dominated the 2013-2016 epidemic. *Cell* **167**, 1088-1098. (doi:10.1016/j.cell.2016.10.014)
13. Walter S, Rasche A, Moreira-Soto A, Pfaender S, Bletsas M, Corman VM, Aguilar-Setien A, García-Lacy F, Hans A, Todt D, et al. 2017 Differential infection patterns and recent evolutionary origins of equine hepaciviruses in donkeys. *J. Virol.* **91**, e01711-16. (doi:10.1128/JVI.01711-16)
14. Quan P-L, Firth C, Conte J, Williams S, Zambranan C, Anthony A, Ellison J, Gilbert A, Kuzmin I, Niezgodna M, et al. 2013 Bats are a major natural reservoir for hepaciviruses

and pegviruses. *Proc. Natl. Acad. Sci. USA* **110**, 8194-8199
(doi:10.1073/pnas.1303037110.)

15. Herfst S, Schrauwen EJ, Linster M, Chutinimitkul S, de Wit E, Munster VJ, Sorrell EM, Bestebroer TM, Burke DF, Smith DJ, et al. 2012 Airborne transmission of influenza A/H5N1 virus between ferrets. *Science* **336**, 1534-1541. (doi:10.1126/science.1213362)

16. Parrish CR, Holmes EC, Morens DM, Park EC, Burke DS, Calisher CH, Laughlin CA, Saif LJ, Daszak P. 2008 Cross-species virus transmission and the emergence of new epidemic diseases. *Microbiol. Mol. Biol. Rev.* **72**, 457-470. (doi:10.1128/mmbr.00004-08)

17. Taubenberger JK, Kash JC. 2010 Influenza virus evolution, host adaptation and pandemic formation. *Cell Host Microbe* **7**, 440-451. (doi:10.1016/j.chom.2010.05.009)

18. Zaraket H, Bridges OA, Russell CJ. 2013 The pH of activation of the hemagglutinin protein regulates H5N1 influenza virus replication and pathogenesis in mice. *J. Virol.* **87**, 4826-4834. (doi:10.1128/jvi.03110-12)

19. Kluge SF, Mack K, Iyer SS, Pujol FM, Heigele A, Learn GH, Usmani SM, Sauter D, Joas S, Hotter D, et al. 2014 Nef proteins of epidemic HIV-1 group O strains antagonize human tetherin. *Cell Host Microbe* **16**, 639-650. (doi: 10.1016/j.chom.2014.10.002)

20. Schindler M, Münch J, Kutsch O, Li H, Santiago ML, Bibollet-Ruche F, Müller-Trutwin MC, Novembre FJ, Peeters M, Courgnaud V, et al. 2006 Nef-mediated suppression of T cell activation was lost in a lentiviral lineage that gave rise to HIV-1. *Cell* **125**, 1055-1067.

21. Crawford PC, Dubovi EJ, Castleman WL, Stephenson I, Gibbs EP, Chen L, Smith C, Hill RC, Ferro P, Pompey J, et al. 2005 Transmission of equine influenza virus to dogs. *Science* **310**, 482-485. (doi:10.1126/science.1117950)

22. Dalziel BD, Huang K, Geoghegan JL, Arinaminpathy N, Dubovi EJ, Grenfell BT, Ellner SP, Holmes EC, Parrish CR. 2014 Contact heterogeneity, rather than transmission efficiency, limits the emergence and spread of canine influenza virus. *PLoS Pathog.* **10**, e1004455. (doi:10.1371/journal.ppat.1004455)

23. Stephen SM. 1995 Factors in the emergence of infectious diseases. *Emerg. Infect. Dis.* **1**, 7-15. (doi:10.3201/eid0101.950102)
24. Johnson CK, Hitchens PL, Evans TS, Goldstein T, Thomas K, Clements A, Joly DO, Wolfe ND, Daszak P, Karesh WB, et al. 2015 Spillover and pandemic properties of zoonotic viruses with high host plasticity. *Sci. Rep.* **5**, 14830. (doi:10.1038/srep14830)
25. Geoghegan JL, Senior AM, Holmes EC. 2016 Pathogen population bottlenecks and adaptive landscapes: overcoming the barriers to disease emergence. *Proc. R. Soc. Lond. B. Biol.* **283**, 20160727. (doi:10.1098/rspb.2016.0727)
26. Lipsitch M, Barclay W, Raman R, Russell CJ, Belser JA, Cobey S, Kasson PM, Lloyd-Smith JO, Maurer-Stroh S, Riley S, et al. 2016 Viral factors in influenza pandemic risk assessment. *eLife* **5**, 18491 (doi:10.7554/eLife.18491)
27. Fonville JM. 2015 Expected effect of deleterious mutations on within-host adaptation of pathogens. *J. Virol.* **89**, 9242-9251. (doi:10.1128/jvi.00832-15)
28. Daszak P, Carroll D, Wolfe N, Mazet J. 2016 The global virome project. *Int. J. Infect. Dis.* **53S**, 4-163. (doi:10.1016/j.ijid.2016.11.097)
29. Shi M, Lin X-D, Tian J-H, Chen L-J, Chen X, Li C-X, Qin X-C, Li J, Cao J-P, Eden J-S, et al. 2016 Redefining the invertebrate RNA virosphere. *Nature* **540**, 539-543. (doi:10.1038/nature20167)
30. Li C-X, Shi M, Tian J-H, Lin X-D, Kang Y-J, Chen L-J, Qin X-C, Xu J, Holmes EC, Zhang Y-Z. 2015 Unprecedented genomic diversity of RNA viruses in arthropods reveals the ancestry of negative-sense RNA viruses. *eLife* **4**, e05378. (doi:10.7554/eLife.05378)
31. Mora C, Tittensor DP, Adl S, Simpson AGB, Worm B. 2011 How many species are there on Earth and in the ocean? *PLoS Biol.* **9**, e1001127. (doi:10.1371/journal.pbio.1001127)
32. Webster CL, Waldron FM, Robertson S, Crowson D, Ferrari G, Quintana JF, Brouqui JM, Bayne EH, Longdon B, Buck AH, et al. 2015 The discovery, distribution, and evolution of viruses associated with *Drosophila melanogaster*. *PLoS Biol.* **13**, e1002210. (doi:10.1371/journal.pbio.1002210)

33. Culley AI, Lang AS, Suttle CA. 2006 Metagenomic analysis of coastal RNA virus communities. *Science* **312**, 1795-1798. (doi:10.1126/science.1127404)
34. Desnues C, Rodriguez-Brito B, Rayhawk S, Kelley S, Tran T, Haynes M, Liu H, Furlan M, Wegley L, Chau B, et al. 2008 Biodiversity and biogeography of phages in modern stromatolites and thrombolites. *Nature* **452**, 340-343. (doi:10.1038/nature06735)
35. Paez-Espino D, Eloë-Fadrosh EA, Pavlopoulos GA, Thomas AD, Huntemann M, Mikhailova N, Rubin E, Ivanova NN, Kyrpides NC. 2016 Uncovering Earth's virome. *Nature* **536**, 425-430. (doi:10.1038/nature19094)
36. Yao Y, Li X, Liao B, Huang L, He P, Wang F, Yang J, Sun H, Zhao Y, Yang J. 2017 Predicting influenza antigenicity from Hemagglutinin sequence data based on a joint random forest method. *Sci. Rep.* **7**, 1545. (doi:10.1038/s41598-017-01699-z)
37. Woolhouse MEJ, Ashworth JL. 2017 Can we identify viruses with pandemic potential? *The Biochemist* **39**, 8-11.
38. García-Sastre A. 2010 Influenza virus receptor specificity: disease and transmission. *Am. J. Pathol.* **176**, 1584-1585. (doi:10.2353/ajpath.2010.100066)
39. Li W, Shi Z, Yu M, Ren W, Smith C, Epstein JH, Wang H, Crameri G, Hu Z, Zhang H, et al. 2005 Bats are natural reservoirs of SARS-like coronaviruses. *Science* **310**, 676-679. (doi:10.1126/science.1118391)
40. Anthony SJ, Johnson CK, Greig DJ, Kramer S, Che X, Wells H, Hicks AL, Joly DO, Wolfe ND, Daszak P, et al. 2017 Global patterns in coronavirus diversity. *Virus Evol.* **3**, vex012-vex012. (doi:10.1093/ve/vex012)
41. Luis AD, Hayman DT, O'Shea TJ, Cryan PM, Gilbert AT, Pulliam JR, Mills JN, Timonin ME, Willis CK, Cunningham AA, et al. 2013 A comparison of bats and rodents as reservoirs of zoonotic viruses: are bats special? *Proc. Biol. Sci.* **280**, 20122753. (doi:10.1098/rspb.2012.2753)
42. Dick GW, Kitchen SF, Haddow AJ. 1952 Zika virus. I. Isolations and serological specificity. *Trans. R. Soc. Trop. Med. Hyg.* **46**, 509-520. (doi:doi.org/10.1016/0035-9203(52)90042-4)

43. Milne-Price S, Miazgowicz KL, Munster VJ. 2014 The emergence of the Middle East Respiratory Syndrome coronavirus (MERS-CoV). *Pathog. Dis.* **71**, 119-134. (doi:10.1111/2049-632X.12166)
44. Schmidt KA, Ostfeld RS. 2001 Biodiversity and the dilution effect in disease ecology. *Ecology* **82**, 609-619. (doi:10.1890/0012-9658(2001)082[0609:BATDEI]2.0.CO;2)
45. Johnson PTJ, Preston DL, Hoverman JT, Richgels KLD. 2013 Biodiversity decreases disease through predictable changes in host community competence. *Nature* **494**, 230-233. (doi:10.1038/nature11883)
46. Allan BF, Langerhans RB, Ryberg WA, Landesman WJ, Griffin NW, Katz RS, Oberle BJ, Schutzenhofer MR, Smyth KN, de St Maurice A, et al. 2009 Ecological correlates of risk and incidence of West Nile virus in the United States. *Oecologia* **158**, 699-708. (doi:10.1007/s00442-008-1169-9)
47. Clay CA, Lehmer EM, Jeor SS, Dearing MD. 2009 Sin Nombre virus and rodent species diversity: a test of the dilution and amplification hypotheses. *PLoS ONE* **4**, e6467. (doi:10.1371/journal.pone.0006467)
48. Haas SE, Hooten MB, Rizzo DM, Meentemeyer RK. 2011 Forest species diversity reduces disease risk in a generalist plant pathogen invasion. *Ecol. Lett.* **14**, 1108-1116. (doi:10.1111/j.1461-0248.2011.01679.x)
49. LoGiudice K, Ostfeld RS, Schmidt KA, Keesing F. 2003 The ecology of infectious disease: effects of host diversity and community composition on Lyme disease risk. *Proc. Natl. Acad. Sci. U.S.A.* **100**, 567-571. (doi:10.1073/pnas.0233733100)
50. Geoghegan JL, Duchêne S, Holmes EC. 2017 Comparative analysis estimates the relative frequencies of co-divergence and cross-species transmission within viral families. *PLoS Pathog.* **13**, e1006215. (doi:10.1371/journal.ppat.1006215)
51. Read AF. 1994 The evolution of virulence. *Trends Microbiol.* **2**, 73-76. (doi:10.1016/0966-842X(94)90537-1)
52. Luksza M, Lassig M. 2014 A predictive fitness model for influenza. *Nature* **507**, 57-61. (doi:10.1038/nature13087)

53. Dill JA, Camus AC, Leary JH, Di Giallonardo F, Holmes EC, Ng TF. 2016 Distinct viral lineages from fish and amphibians reveal the complex evolutionary history of hepadnaviruses. *J. Virol.* **90**, 7920-7933. (doi:10.1128/jvi.00832-16)
54. Katzourakis A, Gifford RJ. 2010 Endogenous viral elements in animal genomes. *PLoS Genet.* **6**, e1001191. (doi:10.1371/journal.pgen.1001191)
55. Lauber C, Seitz S, Mattei S, Suh A, Beck J, Herstein J, Börold J, Salzburger W, Kaderali L, Briggs JAG, et al. 2017 Deciphering the origin and evolution of hepatitis B viruses by means of a family of non-enveloped fish viruses. *Cell Host Microbe* pii S1931-3128 (doi: 10.1016/j.chom.2017.07.019)
56. Horimoto T, Kawaoka Y. 2005 Influenza: lessons from past pandemics, warnings from current incidents. *Nat. Rev. Microbiol.* **3**, 591-600. (doi:10.1038/nrmicro1208)
57. Di Giallonardo F, Holmes EC. 2015 Exploring host-pathogen interactions through biological control. *PLoS Pathog.* **11**, e1004865. (doi:10.1371/journal.ppat.1004865)
58. Moury B, Fabre F, Hébrard E, Froissart R. 2017 Determinants of host species range in plant viruses. *J. Gen. Virol.* **98**, 862-873. (doi:doi:10.1099/jgv.0.000742)
59. Leonard AS, Weissman D, Greenbaum B, Ghedin E, Koelle K. 2017 Transmission bottleneck size estimation from pathogen deep-sequencing data, with an application to human influenza A virus. *J. Virol.* **91**, e00171-17. (doi:10.1128/JVI.00171-17)
60. Zwart MP, Elena SF. 2015 Matters of size: genetic bottlenecks in virus infection and their potential impact on evolution. *Annu Rev Virol* **2**, 161-179. (doi:doi:10.1146/annurev-virology-100114-055135)
61. Grubaugh ND, Fauver JR, Ruckert C, Weger-Lucarelli J, Garcia-Luna S, Murrieta RA, Gendernalik A, Smith DR, Brackney DE, Ebel GD. 2017 Mosquitoes transmit unique West Nile virus populations during each feeding episode. *Cell Rep.* **19**, 709-718. (doi:10.1016/j.celrep.2017.03.076)
62. Stack JC, Murcia PR, Grenfell BT, Wood JL, Holmes EC. 2013 Inferring the inter-host transmission of influenza A virus using patterns of intra-host genetic variation. *Proc Biol Sci* **280**, 20122173. (doi:10.1098/rspb.2012.2173)

63. Moncla LH, Zhong G, Nelson CW, Dinis JM, Mutschler J, Hughes AL, Watanabe T, Kawaoka Y, Friedrich TC. 2016 Selective bottlenecks shape evolutionary pathways taken during mammalian adaptation of a 1918-like avian influenza virus. *Cell Host Microbe* **19**, 169-180. (doi:10.1016/j.chom.2016.01.011)

64. Kelly TR, Karesh WB, Johnson CK, Gilardi KVK, Anthony SJ, Goldstein T, Olson SH, Machalaba C, Mazet JAK. 2017 One Health proof of concept: Bringing a transdisciplinary approach to surveillance for zoonotic viruses at the human-wild animal interface. *Prev. Vet. Med.* **137**, 112-118. (doi:10.1016/j.prevetmed.2016.11.023)